# TRIVLA: A TRIPLE-SYSTEM-BASED UNIFIED VISION-LANGUAGE-ACTION MODEL WITH EPISODIC WORLD MODELING FOR GENERAL ROBOT CONTROL

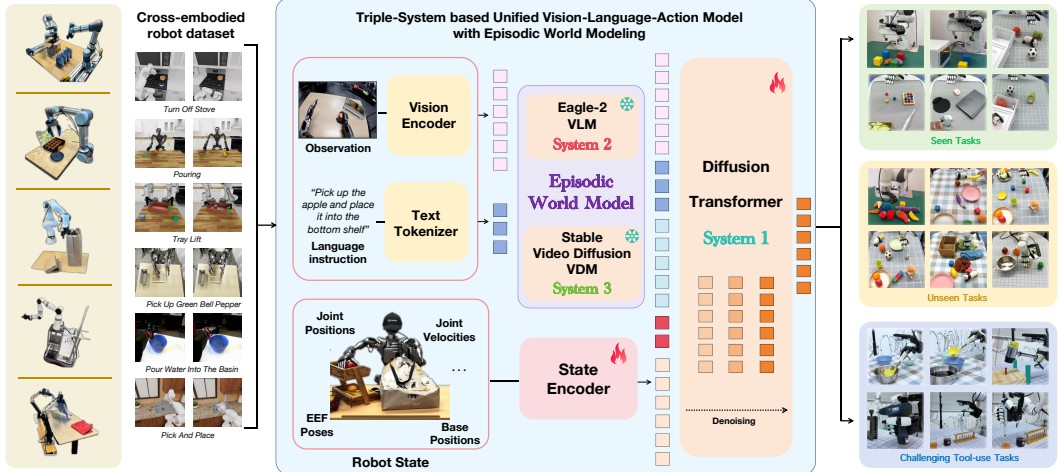

Figure 1: **TriVLA** is a unified Vision-Language-Action framework that adopts a triple-system architecture inspired by the **episodic world model**. Image and language inputs are processed by a Vision-Language Model for multimodal perception. A Video Diffusion Model provides dynamic world modeling and future prediction. The policy module integrates sequential outputs, robot state, and action history and generates real-time actions for complex manipulation tasks.

## ABSTRACT

Recent advances in vision–language models (VLMs) have enabled robots to follow open-ended instructions and demonstrate impressive commonsense reasoning. However, current vision–language–action (VLA) frameworks primarily rely on static representations and limited temporal context, restricting agents to short-horizon, reactive behaviors and hindering robust generalization in dynamic embodied environments. Inspired by cognitive neuroscience theories of episodic memory, we are, to our knowledge, among the first to introduce a formalized episodic world model in VLA, enabling embodied robots to accumulate, recall, and predict sequential experiences. As an instantiation of this concept, our unified **TriVLA** realizes the episodic world model through a triple-system architecture: integrating multimodal grounding from a pretrained VLM (System 2) and temporally rich dynamics perception from a video diffusion model (System 3). This enables the agent to accumulate and recall sequential experiences, interpret current contexts, and predict future environmental evolution. Guided by episodic representations that span both the past and anticipated future, the downstream policy (System 1) generates coherent, context-aware action sequences through flow-matching and cross-modal attention mechanisms. Experimental results show that TriVLA operates efficiently at 36 Hz and consistently outperforms baseline models on standard benchmarks and challenging real-world manipulation tasks. It demonstrates strong long-horizon planning and open-ended intent understanding,

---

[†] Corresponding authors

showcasing the advantages of episodic world model-inspired reasoning for robust, generalizable robot intelligence.

# 1 INTRODUCTION

*"Episodic memory is the only memory system that allows mental time travel—backward into the past and forward into the future."*

— Endel Tulving

Building on this cognitive foundation, we advocate that robotic agents, require an internal **episodic world model**: a representational system that not only recalls past interactions but also anticipates future dynamics, thereby enabling robust generalization in embodied environments.

Decades of cognitive neuroscience provide compelling evidence for this perspective. As first articulated by Tulving Tulving et al. (1972), episodic memory refers to the encoding, storage, and retrieval of experiences within their spatiotemporal context. This unique system empowers humans not only to recollect the past but also to simulate potential futures, thus grounding flexible planning and adaptive decision-making. Converging findings highlight the central roles of the hippocampus and prefrontal cortex in supporting episodic memory, enabling individuals to integrate sensory cues with temporal dynamics to construct predictive internal world models Tulving (2002); Pritzel et al. (2017); Blundell et al. (2016); Lin et al. (2018); Gershman & Daw (2017). Episodic memory thus forms a fundamental component of intelligence, providing both the experiential basis for learning and the representational scaffolding for generalization across novel tasks.

Inspired by these insights, we propose the concept of an episodic world model: a unified framework that integrates multimodal grounding with temporally rich dynamic modeling. Unlike static scene representations, an episodic world model continuously accumulates, recalls, and predicts sequential experiences, equipping artificial agents with a memory system more akin to human intelligence. Recent advances in video diffusion models (VDMs) Blattmann et al. (2023a); Hong et al. (2022); Yang et al. (2024); Brooks et al. (2024) provide a technological foundation for this paradigm, as they capture temporal continuity and physical dynamics across video sequences, naturally aligning with episodic memory principles and enabling richer, context-aware internal representations.

In parallel, vision–language models (VLMs) Liu et al. (2024c); Alayrac et al. (2022); Li et al. (2023a); Zhang et al. (2023); Bai et al. (2023); Gao* et al. (2023); Zhang et al. (2024b;a) have demonstrated impressive progress in instruction following and commonsense reasoning through large-scale pretraining on image–text corpora. Extending these capabilities, dual-system architectures have advanced VLMs into vision–language–action (VLA) models that generate action plans Ahn et al. (2022); Driess et al. (2023); Huang et al. (2023); Belkhale et al. (2024) and estimate SE(3) object poses Brohan et al. (2023); Kim et al. (2024); Li et al. (2024), enabling robots to map multimodal inputs into generalizable control behaviors. As illustrated in Figure 2, current VLM-based VLA systems Intelligence et al. (2025); Black et al. (2024); Brohan et al. (2023); Kim et al. (2024); Pertsch et al. (2025) remain predominantly static: they depend on one or two instantaneous observations, overlooking the sequential and dynamic structures that characterize embodied interaction. As a result, they cannot encode or utilize temporally extended experiences, a capability similar to human episodic memory and crucial for robust performance in dynamic environments.

To bridge this gap, we introduce **TriVLA**, a unified Vision–Language–Action model that implements the episodic world model through a triple-system compositional architecture. Extending prior dual-system designs Bjorck et al. (2025); Shi et al. (2025), TriVLA explicitly integrates:

- System 2: Episodic Multimodal Perception, a pretrained VLM that interprets observations and instructions, summarizing task goals and contextual cues.
- System 3: Episodic Dynamics Perception, a video diffusion model fine-tuned on large-scale human and robotic manipulation datasets Khazatsky et al. (2024); Jin et al. (2024); Lu et al. (2024), which encodes sequences of past states and predicts future scene trajectories, realizing episodic context accumulation.

Together, Systems 2 and 3 jointly compose the episodic world model, fusing descriptive multimodal grounding with predictive temporal modeling. This joint representation empowers Policy Learn-

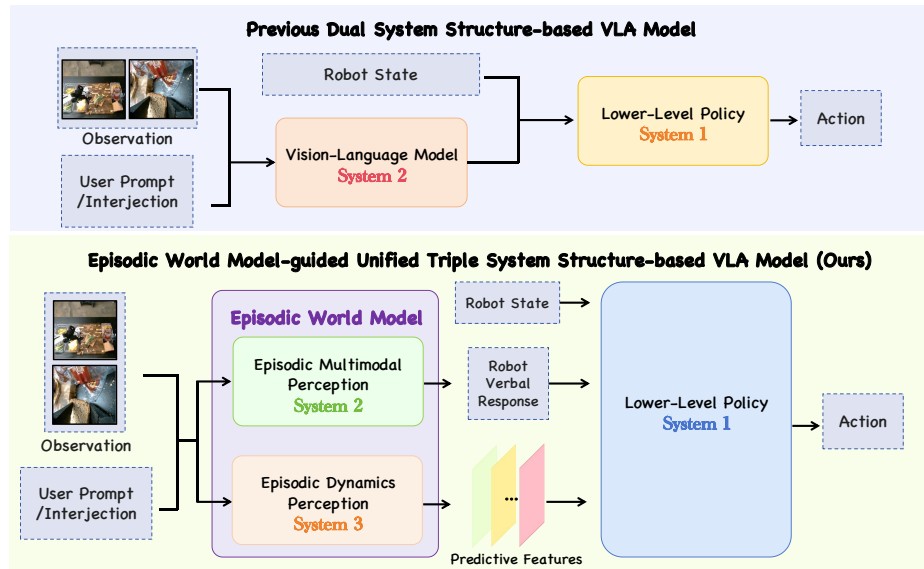

Figure 2: **Comparison between dual-system architectures and our episodic world model-guided TriVLA. TriVLA** implements the **episodic world model** using a triple-system architecture. In contrast, previous dual-system methods relied on static representations and limited temporal context, which restricted agents to short-horizon, reactive behaviors in dynamic environments.

ing (System 1) to flexibly adapt actions based on accumulated experiences and anticipated future dynamics. By continually monitoring motion sequences embedded within the episodic world representation, the downstream policy induces an implicit inverse-dynamics prior Min et al. (2023); Tian et al. (2024). This facilitates the transfer of the generalization capabilities inherent in the episodic world model to the robotic policy, meaning that the robot requires only a limited number of demonstrations to align its action space with the visual domain. During training, System 1 utilizes action flow-matching and cross-attention mechanisms to integrate output tokens from both Systems 2 and 3. It adopts embodiment-specific encoders and decoders to manage variable state and action dimensions during motion generation. In addition, inspired by recent advances in robot learning, System 1 is designed to predict a chunk of actions rather than generating isolated actions at each timestep.

Our main contribution is an episodic world model inspired by cognitive neuroscience theories of episodic memory. Building on this, we introduce a novel triple-system architecture within a unified Vision-Language-Action framework. The episodic world model that guides manipulation policy learning with temporally extended, context-aware signals. As illustrated in Figure 1, this unified framework empowers robots to interpret complex prompts, reason over long horizons, and adapt adaptively in open-ended, dynamic scenarios.

Experimental results demonstrate that the proposed TriVLA consistently outperforms baseline algorithms, in both simulated Mees et al. (2022); Liu et al. (2024a); Yu et al. (2020) and real-world environments. This highlights its effectiveness in aligning with human intent and achieving long-horizon task success. Notably, TriVLA attains improvements of 0.21, 0.11, and 0.13 on the Calvin ABC→D, LIBERO, and MetaWorld benchmarks, respectively, compared to prior state-of-the-art methods. In real-world experiments, TriVLA demonstrates strong effectiveness in dexterous hand manipulation tasks, particularly in long-horizon scenarios.

The contributions of this paper are summarized:

- *Episodic World Model Inspired by Cognitive Neuroscience*: We propose an episodic world model for embodied agents, inspired by cognitive neuroscience theories of episodic memory. This model enables robots to accumulate, recall, and predict sequential multimodal experiences and offers a solid foundation for robust, adaptive control.

- *A Unified Triple-System Compositional Architecture*: Building on this foundation, we present TriVLA, a triple-system architecture implementing the episodic world model.

TriVLA provides high-level reasoning and dynamic prediction. Robots using TriVLA can understand complex prompts and perform long-horizon manipulation.

- *State-of-the-art Performance*: TriVLA outperforms other baseline algorithms, including novel skill compositions beyond training combinations. This demonstrates the effectiveness in both alignment with human intent and long-horizon task success.

## 2 RELATED WORK

**Vision-language-action models.** Previous studies Ahn et al. (2022); Driess et al. (2023); Huang et al. (2023; 2024b) have advanced robotic language-and-vision comprehension to autonomously generate task plans. Vision-language-action (VLA) models leverage VLM reasoning for SE(3) pose prediction: RT2 Brohan et al. (2023) binarizes 7-DoF actions for autoregressive prediction; ManipLLM Li et al. (2024) incorporates affordance priors and chain-of-thought reasoning; Open-VLA Kim et al. (2024) pretrains on Open X-Embodiment O'Neill et al. (2023) for improved generalization; and FAST Pertsch et al. (2025) uses discrete cosine transform for scalable prediction. Cognitively inspired dual-systems such as GR00T N1 Bjorck et al. (2025) and Hi Robot Shi et al. (2025) enhance adaptation to novel scenarios and accelerate task learning. Other VLA approaches Liu et al. (2024d); Huang et al. (2024a); Li et al. (2023b); Wu et al. (2023) achieve continuous action prediction by integrating policy heads (MLP, LSTM Graves & Graves (2012)) with regression losses in imitation learning. Most prior methods are static, relying only on current observations and ignoring sequential dynamics, limiting their ability to encode temporally extended experiences essential for robust performance in dynamic environments. In contrast, TriVLA employs a unified triple-system architecture, enabling robots to interpret complex prompts and perform long-horizon manipulation tasks across diverse scenarios.

**Future prediction in robotics.** Prior studies have explored leveraging future prediction to improve policy learning Bharadhwaj et al. (2024); Chen et al. (2024); Ye et al. (2024); Guo et al. (2024). SuSIE Black et al. (2023) bases its control policy on a predicted future keyframe from InstructPix2Pix Brooks et al. (2023), while UniPi Du et al. (2024) models inverse dynamics across two generated frames. These approaches rely on single-step predictions, which fail to fully capture complex physical dynamics, and denoising predicted images is time-consuming, reducing control frequency. GR-1 Wu et al. (2023) generates future frames and actions autoregressively but produces only one image per forward pass, and its prediction quality lags behind diffusion-based methods. Seer Tian et al. (2024) predicts actions via inverse dynamics on forecasted visual states, and VPP Hu et al. (2024) uses video-model representations for generalist robotic policies. In contrast, our TriVLA integrates Episodic Multimodal Perception and Episodic Dynamics Perception to provide both high-level reasoning and dynamic predictive representations, enabling sequential future frame prediction while maintaining reasoning to guide policy learning. Unlike Seer and other single-step predictors, which lack foundation-model priors and cannot support long-horizon rollout, TriVLA uses a fine-tuned video foundation model to generate high-fidelity multi-step predictions for reliable long-horizon reasoning. TriVLA is not merely an incremental extension of VPP or similar methods; it constructs an instruction-conditioned episodic world model that unifies perception, prediction, and decision-making in a single loop.

## 3 PRELIMINARIES

**Vision-language-action model.** Robotic manipulation remains a core challenge in robotics. Vision-language-action (VLA) models predict a robot's next action—typically the end-effector pose—based on visual observations and human instructions. Recent advances in large pretrained vision-language models (VLMs) have enabled strong generalization across diverse, language-conditioned manipulation tasks. Most VLM-based VLA approaches adopt a dual-system architecture inspired by human cognition Kahneman (2011), supporting higher-level reasoning for complex, long-horizon tasks. At each timestep $t$, the high-level system receives images $\mathbf{o}_t$ from base and wrist-mounted cameras and the open-ended instruction $\mathbf{v}_t^{\text{in}}$ to generate reasoning tokens. The low-level policy combines these tokens with images and robot states to produce an action token sequence $\mathbf{v}_t^{\text{out}} \in \mathcal{V}^n$, where each token represents a discrete bin of one dimension in the robot action space. The final robot action is obtained via a post-processing function $f$, yielding $\mathbf{a}_t = f(\mathbf{v}_t^{\text{out}})$. VLA performs diverse manipula-

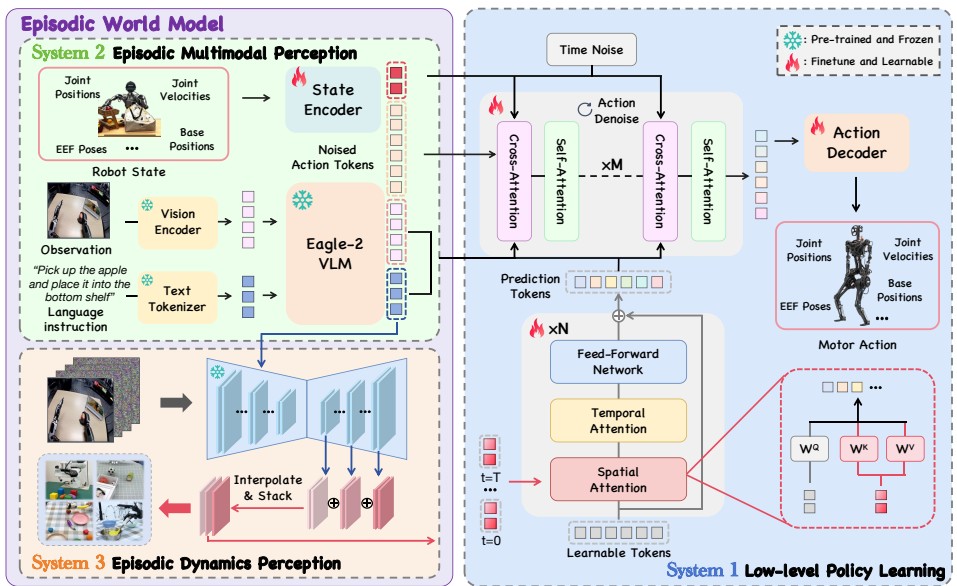

Figure 3: **The pipeline of TriVLA.** TriVLA is a unified Vision-Language-Action framework built on a triple-system paradigm. System 2 employs a pre-trained Eagle-2 VLM for episodic multimodal perception, while System 3 utilizes a general-purpose VDM to model episodic dynamics and sequential changes. Together, these modules form a joint episodic world model with rich, temporally extended representations. System 1 serves as the policy module, applying action flow-matching to integrate all outputs along with robot state and action history.

tion tasks in unstructured, real-world environments by learning generalizable visuomotor behavior from visual observations and language instructions. It is trained via supervised imitation learning on demonstration trajectories.

## 4 OUR PROPOSED TRIVLA

TriVLA implements the episodic world model using a triple-system design (Figure 3). It integrates (i) Episodic Multimodal Perception, which uses the Eagle-2 VLM Li et al. (2025) to interpret visual inputs and language instructions, and (ii) Episodic Dynamics Perception, where a video diffusion model fine-tuned on large-scale manipulation datasets predicts future scene trajectories. These modules provide rich episodic context for policy generalization.

### 4.1 EPISODIC MULTIMODAL PERCEPTION (SYSTEM 2)

To realize this component, TriVLA employs the NVIDIA Eagle-2 vision–language model (VLM) Li et al. (2025), pretrained on large-scale internet corpora to jointly interpret observations and instructions, summarizing task goals and contextual cues. Eagle-2 is built from a SmolLM2 language model and a SigLIP-2 image encoder, aligned through a broad vision–language pretraining protocol. Images are encoded at a resolution of $224 \times 224$ pixels and then processed via pixel shuffle to produce 64 image token embeddings per frame. These image embeddings are jointly encoded with textual input by the LLM component of Eagle-2. The LLM and image encoder are aligned through a general vision-language pretraining protocol covering diverse tasks. During policy training, task descriptions and multiple images are input to the VLM following the chat format established during pretraining. The resulting vision-language tokens, denoted as $Q_{vl}$ (batch size × sequence length × hidden dimension), are extracted from the LLM. Empirically, extracting embeddings from the 12th layer of the LLM, rather than from the final layer, yields faster inference and higher policy success rates. To handle varying robot state dimensions, TriVLA uses an embodiment-specific MLP to project each robot's state into a shared embedding space, resulting in a state token $Q_s$. System 2 (Episodic Multimodal Perception) maintains an online representation of past observations and instructions, capturing relevant context throughout the episode. This representation serves as the model's episodic memory, enabling informed decision-making based on accumulated history.

## 4.2 Episodic Dynamics Perception (System 3)

To infuse extensive prior knowledge of dynamics into policy learning, we fine-tuned the 1.5B-parameter open-source Stable Video Diffusion (SVD) model Blattmann et al. (2023a) as the Episodic Dynamics Perception module for robot manipulation. This video diffusion model, trained on large-scale human and robotic manipulation datasets, encodes sequences of past states and predicts future scene trajectories, enabling episodic context accumulation. By leveraging diverse sources—including internet human manipulation data, robot datasets, and self-collected data—the module robustly models sequential environmental changes essential for effective policy learning. Then Episodic Dynamics Perception module $V_\theta$ is trained with diffusion objective, reconstructing the full video sequence $x_0 = s_{0:T}$ in dataset $D$ from noised samples $x_t = \sqrt{\bar{\alpha}_t} x_0 + \sqrt{1 - \bar{\alpha}_t} \epsilon$:

$$\mathcal{L}_D = \mathbb{E}_{x_0 \sim D, \epsilon, t} \|V_\theta(x_t, l_{emb}, s_0) - x_0\|^2 \tag{1}$$

where $l_{emb}$ denotes the language feature from CLIP Radford et al. (2021).Then we froze the fine-tuned Episodic Dynamics Perception module in downstream action learning.

However, denoising a complete video sequence is computationally intensive and may cause open-loop control problems, as highlighted in Du et al. (2024). Furthermore, videos in raw pixel format frequently contain abundant irrelevant information that can hinder effective decision-making. To mitigate these challenges, we utilize the video diffusion model with a single forward pass. Our key insight is that the initial forward step, despite not producing a clear video, offers a coarse trajectory of future states and informative guidance. This observation is validated experimentally and illustrated in Figure 6. Specifically, we concatenate the current image $s_0$ with the final noised latent $q(x_{t'}|x_0)$ (typically white noise) and input this combination into the System 2. The latent features are then directly utilized. Previous work Xiang et al. (2023) emphasizes that up-sampling layers in diffusion models produce more effective feature representations. The feature at the $m^{th}$ up-sampling layer, with width $W_m$ and height $H_m$, can be expressed as:

$$L_m = V_\theta(x_{t'}, l_{emb}, s_0)_{(m)}, L_m \in \mathbb{R}^{T \times C_m \times W_m \times H_m} \tag{2}$$

To efficiently integrate features from multiple up-sampling layers while eliminating manual layer selection, we propose an automatic feature aggregation approach across layers. First, each layer's feature map is linearly interpolated to a common height and width $W_p \times H_p$:

$$L'_m = \text{Interpolation}(L_m), L'_m \in \mathbb{R}^{T \times C_m \times W_p \times H_p} \tag{3}$$

Subsequently, the features are concatenated along the channel dimension. The final predictive visual representation $F_p \in \mathbb{R}^{T \times (\sum_m C_m) \times W_p \times H_p}$ is given by:

$$F_p = \text{concate}((L'_0, L'_1, \ldots, L'_m), dim = 1)$$

For robots equipped with multiple camera perspectives, including third-person and wrist-mounted cameras, future states are predicted independently for each view, denoted as $F_p^{static}, F_p^{wrist}$. System 3 is not a standalone model. It serves as an *episodic dynamic perception* integrated with Systems 1 and 2. It processes System 2's multimodal representations and extracts multi-layer latent dynamics via cross-layer aggregation. These features provide episodic context for System 1, enabling accurate long-horizon planning. System 3 is the key component that aligns semantics, dynamics, and policy, allowing the model to anticipate future outcomes effectively.

## 4.3 Policy Learning Module (System 1)

Systems 2 and 3 together form the episodic world model, combining descriptive multimodal grounding with predictive temporal modeling. This combined representation allows System 1 (Policy Learning) to flexibly adapt its actions using both accumulated experience and anticipated future dynamics. The predictive representations generated by the video diffusion model remain high-dimensional because they encode sequences of image features. To efficiently aggregate information across spatial, temporal, and multi-view dimensions, TriVLA compresses these representations into a fixed set of tokens. We initializes learnable tokens $Q_{[0:T,0:L]}$ with fixed length $T \times L$, performing spatial-temporal attention Blattmann et al. (2023b) on each corresponding frame, followed by feed-forward layers. Formally, this branch can be expressed as follows where $i$ is the index of frame:

$$Q' = \{\text{Spat-Attn}(Q[i], (F_p^{static}[i], F_p^{wrist}[i]))\}_{i=0}^T$$
$$Q_p = \text{FFN}(\text{Temp-Attn}(Q')) \tag{4}$$

Table 1: **Zero-shot long-horizon evaluation on the Calvin ABC→D benchmark (Avg. Len).**

| Category | Method | Annotated Data | $i^{th}$ Task Success Rate | | | | | |
|---|---|---|---|---|---|---|---|---|
| | | | 1 | 2 | 3 | 4 | 5 | Avg. Len ↑ |
| Direct Action Learning Method | RT-1 | 100%ABC | 0.533 | 0.222 | 0.094 | 0.038 | 0.013 | 0.90 |
| | Diffusion Policy | 100%ABC | 0.402 | 0.123 | 0.026 | 0.008 | 0.00 | 0.56 |
| | Robo-Flamingo | 100%ABC | 0.824 | 0.619 | 0.466 | 0.331 | 0.235 | 2.47 |
| 3D Method | RoboUniview | 100%ABC | 0.942 | 0.842 | 0.734 | 0.622 | 0.507 | 3.65 |
| Future Prediction Related Method | Uni-Pi | 100%ABC | 0.560 | 0.160 | 0.080 | 0.080 | 0.040 | 0.92 |
| | MDT | 100%ABC | 0.631 | 0.429 | 0.247 | 0.151 | 0.091 | 1.55 |
| | Susie | 100%ABC | 0.870 | 0.690 | 0.490 | 0.380 | 0.260 | 2.69 |
| | GR-1 | 100%ABC | 0.854 | 0.712 | 0.596 | 0.497 | 0.401 | 3.06 |
| | Vidman | 100%ABC | 0.915 | 0.764 | 0.682 | 0.592 | 0.467 | 3.42 |
| | Seer | 100%ABC | 0.963 | 0.916 | 0.861 | 0.803 | 0.740 | 4.28 |
| | VPP | 100%ABC | 0.965 | 0.909 | 0.866 | 0.820 | 0.769 | 4.33 |
| | **TriVLA (ours)** | 100%ABC | **0.968** | **0.924** | **0.868** | **0.832** | **0.818** | **4.37** |
| Data Efficiency | GR-1 | 10%ABC | 0.672 | 0.371 | 0.198 | 0.108 | 0.069 | 1.41 |
| | VPP | 10%ABC | 0.878 | 0.746 | 0.632 | 0.540 | 0.453 | 3.25 |
| | **TriVLA (ours)** | 10%ABC | **0.914** | **0.768** | **0.644** | **0.564** | **0.512** | **3.46** |

Table 2: **LIBERO benchmark experimental results.**

| | Average (↑) | Spatial (↑) | Object (↑) | Goal (↑) | Long (↑) |
|---|---|---|---|---|---|
| Diffusion Policy | $72.4 \pm 0.7\%$ | $78.3 \pm 1.1\%$ | $92.5 \pm 0.7\%$ | $68.3 \pm 1.2\%$ | $50.5 \pm 1.3\%$ |
| Octo | $75.1 \pm 0.6\%$ | $78.9 \pm 1.0\%$ | $85.7 \pm 0.9\%$ | $84.6 \pm 0.9\%$ | $51.1 \pm 1.3\%$ |
| OpenVLA | $76.5 \pm 0.6\%$ | $84.7 \pm 0.9\%$ | $88.4 \pm 0.8\%$ | $79.2 \pm 1.0\%$ | $53.7 \pm 1.3\%$ |
| TriVLA (ours) | $\mathbf{87.0 \pm 0.7\ \%}$ | $\mathbf{91.2 \pm 0.8\%}$ | $\mathbf{93.8 \pm 0.7\%}$ | $\mathbf{89.8 \pm 0.9\%}$ | $\mathbf{73.2 \pm 0.5\%}$ |

After the Episodic Multimodal Perception module (System 2) extracts vision-language tokens $Q_{vl}$, and the Episodic Dynamics Perception module (System 3) aggregates future dynamic features into predictive tokens $Q_p$, a diffusion policy is employed as the action head to generate the action sequence $a_0 \in A$ conditioned on $Q_{vl}$ and $Q_p$. The aggregated tokens $Q_{vl}$ and $Q_p$ are integrated into the diffusion transformer blocks via cross-attention layers. The diffusion policy aims to reconstruct the original action $a_0$ from the noised action $a_k = \sqrt{\bar{\beta}_k}a_0 + \sqrt{1 - \bar{\beta}_k}\epsilon$, where $\epsilon$ denotes white noise and $\bar{\beta}_k$ is the noise coefficient at step $k$. This process can be interpreted as learning a denoiser $D_\psi$ to approximate the noise $\epsilon$. After the final DiT block, we apply an embodiment-specific action decoder $A_d$, implemented as a multi-layer perceptron (MLP). This decoder processes the final tokens to predict actions and minimize the following loss function:

$$\mathcal{L}_{\text{diff}}(\psi; A) = \mathbb{E}_{a_0, \epsilon, k} \|A_d(D_\psi(a_k, Q_{vl}, Q_p)) - a_0\|^2 \tag{5}$$

## 5 EXPERIMENTS

**Simulation Benchmarks.** We evaluate our method on three widely used simulation benchmarks for long-horizon manipulation. CALVIN Mees et al. (2022) assesses policies in the challenging ABC→D generalization setting; following GR-1 Wu et al. (2023), we use only language-annotated ABC data for training and testing in the unseen D environment. LIBERO Liu et al. (2024a) consists of four suites (Spatial, Object, Goal, Long), each with 10 tasks and 50 demonstrations. Meta-World Yu et al. (2020); Radosavovic et al. (2023) includes 50 Sawyer-robot tasks.

**Real-world Experimental Setups.** We use a KINOVA GEN2 robot with a RealSense D455 depth camera mounted in an eye-to-hand configuration. In an indoor environment, we arranged various objects to encourage generalization of manipulation skills across different scenes. To further evaluate the episodic world model, we designed long-horizon, high-dynamic tasks that require the agent to accumulate, recall, and predict sequential multi-modal experiences. The RealSense D455 captured the entire scene and the robot's state from both third-person and wrist perspectives.

### 5.1 EXPERIMENTAL RESULTS

**Quantitative Results.** Comparisons on the CALVIN benchmark are presented in Table 1. Zero-shot long-horizon evaluation (CALVIN ABC→D) tests the policy on held-out tasks without fine-tuning.

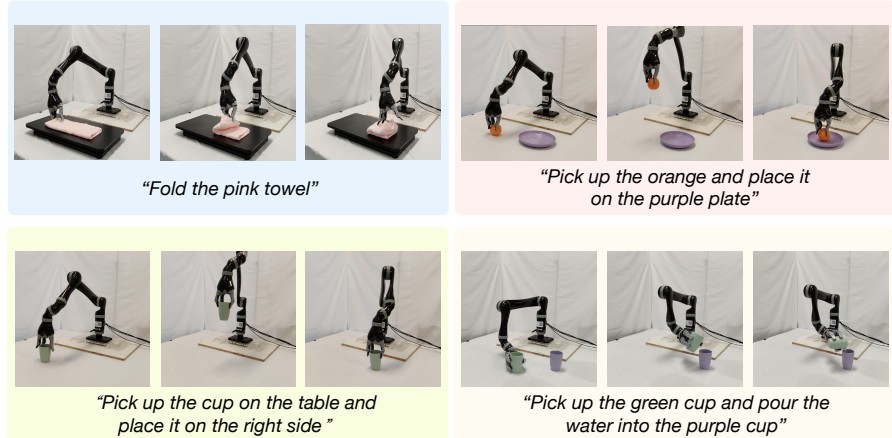

Figure 4: **Qualitative case study of short-horizon tasks.** Our **TriVLA** performs well on short-horizon tasks. In the real-world tasks, it leverages a triple-system architecture that unifies Episodic Multimodal Perception and Dynamics Perception—both crucial for generalizable policy learning.

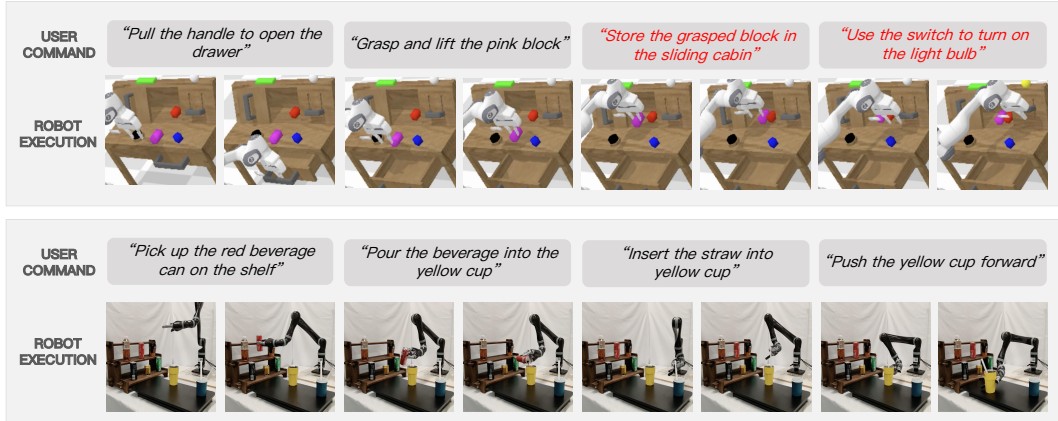

Figure 5: **Qualitative results of long-horizon tasks.** Our **TriVLA** performs well on long-horizon tasks. In the CALVIN and real-world tasks, it leverages a triple-system architecture that unifies multiple systems for generalizable policy learning.

Experimental results for Robo-Flamingo, Susie, GR-1, and 3D Diffuser Actors are taken from their original publications. MDT results come from its official implementation, while RT-1 and UniPi results are from Li et al. (2023b) and Black et al. (2023), respectively. We also evaluated Diffusion Policy using the official open-source code with CLIP-based language conditioning. Our proposed TriVLA significantly outperforms previous state-of-the-art methods. Remarkably, trained on only 10% of the annotated CALVIN ABC dataset, TriVLA achieves an average task completion length of 3.46, surpassing methods trained on the full dataset. It also attains the highest performance on the MetaWorld benchmark (60 tasks; Table 5) and exceeds the strongest VPP baseline in average success rate. Quantitative results on LIBERO are shown in Table 2, with each method evaluated over 500 trials per task suite using 3 random seeds.

The results demonstrate that TriVLA effectively adapts to LIBERO simulation tasks, achieving the best or competitive performance.

**Real-World Quantitative Results** TriVLA achieves the highest success rates across four real-world tasks (overall 93.5%; Table 3), outperforming Diffusion Policy, Seer, and VPP. While System 2's multimodal grounding provides modest gains on simple CALVIN tasks, it enables TriVLA to significantly outperform VPP on complex, multi-step instructions like "pre-

Table 5: **Performance on the MetaWorld.**

| Method | Easy | Middle | Hard | Avg ↑ |
|---|---|---|---|---|
| RT-1 | 0.603 | 0.030 | 0.014 | 0.331 |
| Diffusion Policy | 0.433 | 0.072 | 0.089 | 0.299 |
| Susie | 0.542 | 0.213 | 0.244 | 0.420 |
| GR-1 | 0.695 | 0.337 | 0.448 | 0.582 |
| VPP | 0.822 | 0.507 | 0.519 | 0.679 |
| **TriVLA (ours)** | **0.857** | **0.528** | **0.563** | **0.714** |

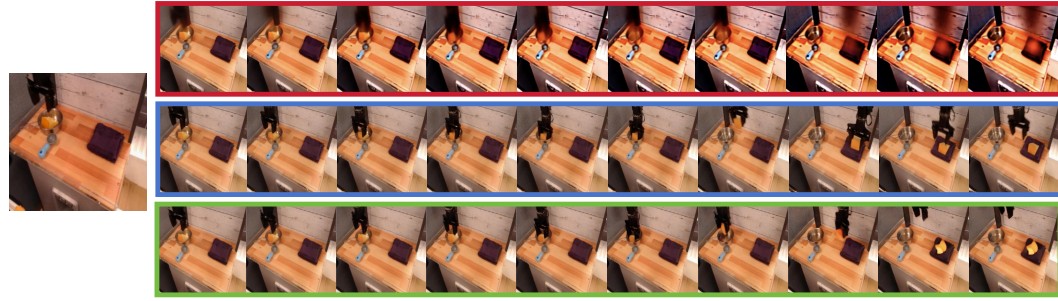

*"Put the cheese onto the cloth"*

Figure 6: **Visualization of Episodic Dynamics Perception.** The red box indicates one-step prediction, the blue box corresponds to full-step prediction, and the green box marks the ground truth.

Table 3: **Success rates (%) on the Real-World Experiment. We compare our TriVLA with baselines on the real-world experiment. The baselines include Diffusion Policy, Seer, and VPP.**

| Method | Fold Towel | Pick up Oranges | Grasp the Cup | Pouring | Overall |
|--------|-----------|----------------|--------------|---------|---------|
| Diffusion Policy | $34\pm_3$ | $42\pm_4$ | $38\pm_4$ | $32\pm_5$ | 36.5 |
| Seer | $66\pm_4$ | $72\pm_6$ | $55\pm_5$ | $66\pm_8$ | 64.8 |
| VPP | $85\pm_4$ | $82\pm_8$ | $78\pm_6$ | $72\pm_2$ | 79.3 |
| **TriVLA (ours)** | $96\pm_4$ | $98\pm_2$ | $89\pm_3$ | $91\pm_4$ | 93.5 |

Table 4: **Success rates (%) on the Real-World Experiment. Sequence of actions: *pick up can → pour into cup → insert straw → push cup*. Baselines include Diffusion Policy, Seer, and VPP.**

| Method | Pick up can | Pour into cup | Insert straw | Push cup | Overall |
|--------|------------|--------------|-------------|----------|---------|
| VPP | $90\pm_3$ | $73\pm_2$ | $64\pm_6$ | $60\pm_3$ | 71.8 |
| UP-VLA | $96\pm_2$ | $92\pm_3$ | $87\pm_1$ | $85\pm_4$ | 90.0 |
| F1 | $96\pm_3$ | $93\pm_2$ | $89\pm_6$ | $85\pm_3$ | 90.8 |
| **TriVLA (ours)** | $98\pm_2$ | $97\pm_3$ | $94\pm_5$ | $91\pm_2$ | **95.0** |

pare a drink for a customer" (95.0 vs. 71.8; Table 4) and surpass concurrent VLM–video architectures such as UP-VLA and F1 (95.0 vs. 90.0/90.8), thanks to the integrated design of Systems 1–3 that supports multimodal grounding, episodic recall, and accurate future prediction for long-horizon task execution.

**Qualitative Results.** Figure 4 and 5 shows two qualitative examples of action sequences in both simulation and the real world. Given multiple consecutive instructions, TriVLA can comprehend intent and leverage prediction to complete long-horizon tasks. These results show that TriVLA supports generalizable policy learning by integrating Episodic Multimodal Perception and Episodic Dynamics Perception. Robots using TriVLA can understand complex sequential prompts, and reason across extended event horizons.

## 5.2 ABLATION STUDY

**Visualization of Episodic Dynamics Perception.** We use a stable video diffusion model as the Episodic Dynamics Perception (EDP) module, whose forward pass produces representations capturing both the current scene context and long-horizon future dynamics. Figure 6 visualizes ground-truth futures, single-step predictions, and full-sequence predictions on the Bridge benchmark Walke et al. (2023). While full-sequence predictions remain reasonable, single-step outputs emphasize key motion cues—such as object and robot-arm movements—that support downstream action learning. Overall, the module effectively models entire video sequences and predicts future frames conditioned on current observations and instructions.

**Effectiveness of the Episodic Multimodal Perception Module.** The System 2 Episodic Multimodal Perception module (EMP), a pretrained Vision-Language Model (VLM), processes visual observations and language instructions to infer task goals. As shown in Table 6 integrating EMP improves performance from 4.06 to 4.37, with inference time increasing from 136 ms to 155 ms, indicating that System 2 substantially enhances action generation accuracy. Similarly, on LIBERO, the inclusion of EMP increases the average success rate from 0.800 to 0.870 (Table 7), demonstrating

Table 6: **Sub-system ablation studies on the CALVIN.**

| EMP | EDP | L-Policy | Task Success Rate ↑ | | | | | Avg.Len ↑ | Latency ↓ | Params ↓ |
|---|---|---|---|---|---|---|---|---|---|---|
| | | | 1 | 2 | 3 | 4 | 5 | | | |
| | | ✓ | 0.914 | 0.772 | 0.703 | 0.622 | 0.511 | 3.68 | 29.29ms | 0.53B |
| ✓ | | ✓ | 0.942 | 0.902 | 0.843 | 0.781 | 0.713 | 4.04 | 59.42ms | 2.07B |
| | ✓ | ✓ | 0.928 | 0.896 | 0.855 | 0.792 | 0.705 | 4.06 | 115.19ms | 1.87B |
| ✓ | ✓ | ✓ | **0.968** | **0.924** | **0.868** | **0.832** | **0.818** | **4.37** | **142.69ms** | **3.39B** |

Table 7: **Sub-system ablation studies on the LIBERO.**

| EMP | EDP | L-Policy | Task Success Rate ↑ | | | | Avg.SR ↑ | Latency ↓ | Params ↓ |
|---|---|---|---|---|---|---|---|---|---|
| | | | 1 | 2 | 3 | 4 | | | |
| | | ✓ | 0.728 | 0.793 | 0.744 | 0.529 | 0.698 | 30.12ms | 0.53B |
| ✓ | | ✓ | 0.813 | 0.852 | 0.883 | 0.682 | 0.808 | 58.44ms | 2.07B |
| | ✓ | ✓ | 0.822 | 0.846 | 0.865 | 0.668 | 0.800 | 118.27ms | 1.87B |
| ✓ | ✓ | ✓ | **0.912** | **0.938** | **0.898** | **0.732** | **0.870** | **141.58ms** | **3.39B** |

Table 8: **Ablation of Stable Video Diffusion (SVD) in System 3.**

| | 1 (↑) | 2 (↑) | 3 (↑) | 4 (↑) | 5 (↑) | Avg.Length(↑) |
|---|---|---|---|---|---|---|
| TriVLA (pretrained SVD) | 0.931 | 0.884 | 0.845 | 0.776 | 0.702 | 3.96 |
| TriVLA (finetuned SVD) | **0.968** | **0.924** | **0.868** | **0.832** | **0.818** | **4.37** |

Table 9: **Ablation of Predicted Step in System 3.**

| | 1 (↑) | 2 (↑) | 3 (↑) | 4 (↑) | 5 (↑) | Avg.Length(↑) |
|---|---|---|---|---|---|---|
| TriVLA (1-step) | 0.930 | 0.866 | 0.842 | 0.744 | 0.708 | 3.94 |
| TriVLA (2-step) | 0.935 | 0.877 | 0.841 | 0.782 | 0.737 | 4.08 |
| TriVLA (4-step) | 0.943 | 0.881 | 0.840 | 0.784 | 0.758 | 4.16 |
| TriVLA (8-step) | 0.944 | 0.892 | 0.854 | 0.792 | 0.774 | 4.22 |
| TriVLA (16-step) | 0.968 | 0.924 | 0.868 | 0.832 | 0.818 | 4.37 |
| TriVLA (32-step) | 0.962 | 0.918 | 0.872 | 0.844 | 0.820 | 4.35 |

that System 2 consistently provides significant gains. In the "L-Policy + EDP" ablation, System 2 (EMP) is removed, so instructions are fed directly into EDP via CLIP. Without EMP's multimodal perception, instruction grounding is weaker, limiting performance on complex tasks.

**Effectiveness of the Episodic Dynamics Perception Module.** The System 3 Episodic Dynamics Perception module (EDP) is a video diffusion model fine-tuned on diverse human and robot manipulation datasets to enhance predictive modeling. We ablate System 3 (EDP) by comparing TriVLA variants on CALVIN and LIBERO. On CALVIN, adding EDP to EMP + L-Policy increases average trajectory length from 4.04 to 4.37 across all five tasks (Table 6). On LIBERO, incorporating EDP improves average success rate from 0.808 to 0.870 (Table 7), demonstrating consistent gains and confirming that the full Triple-System (VLM + EDP + Policy) outperforms the Dual-System (VLM + Policy) in long-horizon, dynamic manipulation tasks. Further ablations show that fine-tuning the Stable Video Diffusion (SVD) model is essential: TriVLA with finetuned SVD outperforms pretrained SVD across all five CALVIN tasks (Avg. Length 4.37 vs. 3.96; Table 8). Horizon ablation also confirms that longer multi-step rollouts consistently improve performance (Avg. Len 3.94 → 4.37), highlighting that effective long-horizon manipulation requires true multi-step dynamics modeling rather than trivial single-step extension.

## 6 CONCLUSION

TriVLA is the first framework to formalize an episodic world model within a unified triple-system architecture, drawing inspiration from cognitive neuroscience theories of episodic memory. By integrating multimodal grounding and rich temporal dynamics, TriVLA provides high-level reasoning and dynamic prediction, enabling robots to accumulate, recall, and predict sequential experiences. Experiments show that TriVLA operates efficiently and consistently outperforms state-of-the-art policy baselines. TriVLA significantly improves long-horizon reasoning, sample efficiency, and open-ended goal achievement. These results highlight the potential of episodic world model reasoning as a solid foundation for robust and generalizable robot control systems.

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

## SUPPLEMENTAL MATERIAL

To better understanding of this work, we offer additional details, analysis, and results as follows:

- **A    Use of Large Language Models, Ethics Statement, and Reproducibility Statement**
  In this section, we clarify the use of LLMs, the ethics statement, and the reproducibility statement in our paper, which primarily served as assistants for refining and polishing the paper during the writing process.

- **B    Implementation Details**
  In this section, we present the implementation details of TriVLA and its inherent Episodic World Model, including the training procedure and rollout process.

- **C    Demo Video**
  In this section, we present the performance of TriVLA on short-horizon and long-horizon tasks to verify the practical effect of the Episodic World Model.

- **D    Comparison Methods**
  In this section, we select a representative subset of prior methods for comparison.

- **E    Details and More Results of Episodic Dynamics Perception**
  In this section, we present detailed visualizations and results for Episodic Dynamics Perception in TriVLA. We employ a stable video diffusion model as the core module for Episodic Dynamics Perception and visualize the intermediate predictive representations through one-step and full step predictions.

- **F    Qualitative Analysis and Results.**
  This section presents comprehensive experiments on simulated and real-world tasks to evaluate the TriVLA framework. The simulated experiments employ three benchmarks that encompass diverse robot embodiments and manipulation tasks. In parallel, real-world trials evaluate long-horizon tabletop manipulation using a Kinova Gen3 robotic arm.

- **G    Real-world Experiments**
  In this section, we present a series of real-world experiments designed to rigorously evaluate the practical applicability, task generalization, and operational robustness of our TriVLA framework under realistic and unstructured environments.

## A    USE OF LARGE LANGUAGE MODELS, ETHICS STATEMENT, AND REPRODUCIBILITY STATEMENT

### A.1    USE OF LARGE LANGUAGE MODELS (LLMS)

In preparing this paper, we used ChatGPT-4o (OpenAI) as a general-purpose writing assistance tool. Its role was strictly limited to checking and improving spelling, grammar, and sentence-level clarity. The LLM did not contribute to the conception of the research idea, experimental design, data analysis, interpretation of results, or the drafting of any substantive scientific content. All intellectual contributions, arguments, and conclusions presented in this paper are our own.

### A.2    ETHICS STATEMENT

TriVLA advances vision–language–action (VLA) research by introducing an episodic world model that enables embodied agents to accumulate, recall, and predict sequential experiences for robust, long-horizon decision-making. This capability lowers barriers to building more generalizable and intelligent robots, which can benefit applications in assistive robotics, manufacturing, and human–robot interaction. At the same time, these advances raise ethical considerations regarding potential misuse (e.g., autonomous systems acting beyond intended safety boundaries, reinforcement of social or cultural biases in pretrained vision–language models) and broader societal impacts (e.g., displacement of human labor or over-reliance on autonomous decision-making). We encourage further research into transparent evaluation, safety alignment, and bias mitigation, as well as careful consideration of the ethical implications when deploying such systems in real-world settings.

### A.3 Reproducibility Statement

We are committed to ensuring the reproducibility of all results reported in this work. Upon publication, we will release the TriVLA codebase, pretrained checkpoints for all system components (vision–language model, video diffusion model, and policy network), as well as training and evaluation scripts. Detailed descriptions of the episodic world model architecture, triple-system integration, and flow-matching mechanisms are provided in Supplemental Material B, along with the hyperparameters and dataset preprocessing steps. We also specify all benchmarks, metrics, and experimental protocols used in both simulated and real-world evaluations. To facilitate independent verification, we will include environment setup instructions, hardware requirements, and example configurations, ensuring the seamless replication of our results.

## B Implementation details

### B.1 Training Details.

As detailed in Section 4, we employ a unified triple-system architecture. System 2, the Episodic Multimodal Perception, utilizes the pretrained Eagle-2 VLM for processing visual and language inputs on an NVIDIA H100 GPU. System 3 fine-tunes a video foundation model for manipulation-centric Episodic Dynamics Perception using 193,690 human Goyal et al. (2017) and 179,074 robotic O'Neill et al. (2023) trajectories, supplemented by videos from CALVIN ABC, MetaWorld, and real-world tasks. To mitigate dataset discrepancies, we adopt dataset-specific sampling ratios following Octo. Video model fine-tuning requires 2–3 days on 8 NVIDIA H100 GPUs. The generalist policy is subsequently trained on task datasets, taking 5–9 hours on four H100 GPUs.

### B.2 Roll-out Details.

The System 2 Episodic Multimodal Perception module employs a pretrained Eagle-2 VLM to extract vision-language tokens, operating at 36.36 Hz on an NVIDIA H100 GPU. In contrast to prior methods, which utilize computationally intensive video denoising—resulting in low control frequenciesBlack et al. (2023) or open-loop limitationsDu et al. (2024)—our approach processes each observation through System 2 only once, during the initial forward pass of the Episodic Dynamics Perception module, with inference latency below 85.9 ms. Subsequently, the downstream policy generates a 10-step action chunk Chi et al. (2023), enabling control frequencies of 34–36 Hz on a consumer-grade NVIDIA RTX H100 GPU.

## C Demo Video

The attached video demonstrates the application of TriVLA, a triple-system architecture inspired by cognitive neuroscience, designed to enhance the capabilities of embodied agents in complex tasks. TriVLA integrates an episodic world model, enabling robots to accumulate, recall, and predict sequential multimodal experiences. This model provides the foundation for robust, adaptive control by simulating episodic memory processes. This showcases the generalization ability of the TriVLA framework. The tasks in the video highlight how TriVLA's high-level reasoning and dynamic prediction enable robots to handle long-horizon manipulation and understand complex prompts, demonstrating its capability for sophisticated, adaptable decision-making.

The following is a detailed explanation of the tasks that TriVLA handles in the video.

### C.1 Visualization of Episodic Dynamics Perception

This demo presents the Episodic Dynamics Perception capability of TriVLA in real-world scenarios. By utilizing a stable video diffusion model, TriVLA encodes the current scene and anticipates future dynamics over long time horizons. The visualization highlights ground-truth outcomes, single-step predictions, and full-sequence forecasts, demonstrating how the model captures essential motion cues such as object interactions and robotic arm trajectories. These results show that TriVLA can effectively model entire video sequences and predict future states based on current observations and

task instructions, enabling a deeper understanding of physical dynamics for downstream decision-making and action planning.

## C.2 VISUALIZATION OF ACTION TRAJECTORY IN SIMULATION

This demo illustrates the action trajectory generation of TriVLA in a simulated environment. We showcase three representative examples of action sequences executed under multiple consecutive instructions, such as "Pull the handle to open the drawer," "Grasp and lift the pink block," "Use the switch to turn on the light bulb," and "Store the grasped block in the sliding cabin." TriVLA demonstrates its ability to comprehend complex instructions, infer underlying intent, and leverage predictive modeling to plan and execute long-horizon tasks. By combining high-level reasoning from vision-language models (VLMs) with dynamic predictive representations from video diffusion models (VDMs), TriVLA integrates world knowledge to enhance intent understanding and predicts future states to guide sequential decision-making. These results highlight how TriVLA effectively coordinates perception, reasoning, and prediction to accomplish complex, multi-step tasks in simulation.

## C.3 VISUALIZATION OF ACTION TRAJECTORY IN SHORT-HORIZON REAL-WORLD TASKS

Across the four short-horizon tasks—folding a pink towel, grasping an orange and placing it on a purple plate, pouring water from a green cup into a purple cup, and relocating a cup to the right side—the TriVLA consistently demonstrates its capability for precise, reliable, and context-aware manipulation. These tasks collectively highlight the model's versatility: from handling deformable objects to executing accurate pick-and-place operations and controlling dynamic pouring actions. By integrating real-time visual perception with adaptive motor planning, TriVLA achieves robust short-horizon performance, ensuring accurate execution under diverse manipulation challenges. This suite of tasks underscores TriVLA's efficiency in short-term control, where rapid perception-action coupling is critical for success.

**Short-horizon Task: "Fold Towel"** In this scenario, the TriVLA demonstrates its capacity to manipulate deformable objects by folding a pink towel with precision. This task requires careful handling of soft materials, demanding spatial reasoning beyond rigid object grasping. The policy leverages its perception to recognize the towel's shape and orientation, planning an appropriate folding trajectory. Through this task, TriVLA highlights its short-horizon ability to interact with deformable objects in a controlled manner.

- TriVLA achieves this by adjusting its grasp and fold strategy in real-time, ensuring the towel is folded along the intended line.
- The success of this task emphasizes TriVLA's adaptability in dealing with non-rigid objects, integrating visual cues into consistent action sequences.

**Short-horizon Task: "Grasp Orange"** In this scenario, the TriVLA showcases its skill in precise object manipulation by picking up an orange and placing it onto a purple plate. The task requires accurate localization and trajectory control, as the system must not only grasp the fruit securely but also transport it safely to the designated location. Through this task, TriVLA demonstrates its ability to carry out reliable short-horizon pick-and-place operations.

- TriVLA accomplishes this by dynamically refining its grasp and movement trajectory based on real-time feedback.
- The successful completion highlights TriVLA's integration of perception and motion planning, enabling robust execution of targeted placement tasks.

**Short-horizon Task: "Pouring"** In this scenario, the TriVLA demonstrates its ability to perform liquid transfer by grasping a green cup and pouring water into a purple cup. This task requires stable control of orientation and precise alignment between the two containers, ensuring minimal spillage. The policy leverages its multimodal perception to model both rigid object states and the flow dynamics of the liquid. Through this task, TriVLA demonstrates competence in controlled pouring, a challenging short-horizon manipulation.

- TriVLA achieves this by continuously monitoring the cup angle and relative position to regulate water flow.

- The success of this task highlights TriVLA's ability to coordinate fine-grained motor control with visual guidance for dynamic object interaction.

**Short-horizon Task: "Pick Cup"** In this scenario, the TriVLA performs a straightforward relocation task by picking up a cup from the table and placing it on the right side. While simple, this task demands accurate detection of the cup's position and a reliable transfer motion without disrupting the environment. Through this task, TriVLA showcases its efficiency in executing basic, short-horizon manipulation.

- TriVLA achieves this by generating a direct grasp-to-place trajectory, adapting its motion based on sensory feedback.

- The success of this task demonstrates TriVLA's ability to reliably execute fundamental pick-and-place operations with consistency.

### C.4 VISUALIZATION OF ACTION TRAJECTORY IN LONG-HORIZON REAL-WORLD TASKS

**Long-horizon Task: "Beverage Preparation"** In this scenario, the TriVLA demonstrates its competence in executing a complex, sequential task: picking up a red beverage can from the shelf, pouring the beverage into a yellow cup, inserting a straw into the yellow cup, and finally pushing the cup forward. Unlike short-horizon manipulations, this task requires the integration of multiple atomic actions into a coherent sequence, demanding sustained spatial reasoning, memory of intermediate states, and precise coordination across distinct phases. Through this task, TriVLA showcases its ability to plan and execute long-horizon activities where success depends on maintaining consistency across multiple dependent steps.

- TriVLA achieves this by decomposing the task into sub-goals, dynamically adjusting its strategy based on real-time perception and the evolving state of the environment.

- The successful completion highlights TriVLA's ability to combine high-level planning with fine-grained control, ensuring the transition is seamless and reliable.

- This task demonstrates TriVLA's strength in long-horizon reasoning, where sustained action sequences and contextual understanding are essential for achieving complex goals.

TriVLA's episodic world model enables robots to simulate memory processes, allowing them to accumulate sequential experiences and predict future actions. This capability helps robots adapt dynamically to changing environments and illustrates how embodied agents can reason about actions and experiences in a human-like way. It adopts a triple-system architecture that integrates episodic memory, high-level reasoning, and dynamic prediction. This unified design allows robots to understand multi-step tasks, solve complex manipulation problems, and make decisions grounded in both past experiences and anticipated outcomes.

In summary, TriVLA provides a robust framework for robots, offering spatial-temporal awareness, high-level reasoning, and adaptive control over long horizons. The model demonstrates exceptional generalization ability, enabling robots to perform tasks in diverse, complex environments.

## D COMPARISON METHODS

Generalist robot policies have been extensively investigated in prior research. In our experiments, we select a representative subset of prior methods for comparison, focusing on those that have achieved state-of-the-art performance or employ approaches similar to ours.

- RT-1 Brohan et al. (2022): A general action learning robot policy integrating semantic features via Efficient-Net with FiLM-conditioning, subsequently employing token learners.

- Diffusion Policy Chi et al. (2023): A action learning approach modeling the robot's visuo-motor policy as a conditional denoising diffusion process enhanced with action diffusers.

- Robo-Flamingo Li et al. (2023b): A direct action learning policy leveraging a pre-trained LLM, integrating visual information into each layer following the Flamingo Alayrac et al. (2022).

- UniPi Du et al. (2024): Initiates by training a video prediction model for future sequence generation and concurrently learns an inverse kinematics model between frames to infer actions.

- MDT Reuss et al. (2024): Trains a diffusion transformer-based policy complemented by an auxiliary MAE loss to facilitate future state reconstruction.

- Susie Black et al. (2023): Employs a fine-tuned InstructPix2Pix Brooks et al. (2023) model to generate goal images and trains a downstream diffusion policy conditioned on these goal images.

- GR-1 Wu et al. (2023): Models video and action sequences using an autoregressive transformer. During policy execution, GR-1 predicts one future frame followed by a action.

- Robo-Uniview Liu et al. (2024b): Develops a 3D-aware visual encoder supervised by a 3D occupancy loss for policy learning.

- Vidman Wen et al. (2024): Pre-trained on the Open X-Embodiment video dataset, it employs a self-attention adapter to convert video representations into policies. However, Vidman's performance is suboptimal due to the absence of fine-tuning the video model on downstream tasks.

- Seer Tian et al. (2024): Designs a novel end-to-end framework that leverages predictive inverse dynamics models to integrate vision and action for scalable robotic manipulation.

- VPP Hu et al. (2024): Leverages video diffusion models to generate visual representations, addressing the limitations of traditional vision encoders in capturing temporal aspects critical for robotic manipulation.

## E  DETAILS AND MORE RESULTS OF EPISODIC DYNAMICS PERCEPTION

We employ a stable video diffusion model as the dynamics perception module, performing a single forward pass to obtain visual representations that encompass both current static information and predicted future dynamics. As illustrated in Figure 7, we present visualizations of ground-truth futures alongside single-step and full-step predictions on the Bridge benchmark. The visualization results indicate that single-step representations convey critical information, including object and robot arm motion, thereby effectively supporting downstream action learning. The dynamics perception module models entire video sequences and predicts future frames conditioned on current observations and instructions, demonstrating a sound understanding of physical dynamics.

Additionally, we extend our analysis to real-world experiments, where we replicate the same predictive framework on an actual robot. As shown in Figure 8 and 9, the true-to-life experimental results mirror the findings from the simulated setup, further validating the robustness of our prediction model. These real-world predictions are crucial, as they show that our system not only captures the motion of objects and the robot arm but also adapts to real-world uncertainties, such as sensor noise and minor mechanical inaccuracies.

The real-world prediction visualizations highlight several important aspects of our approach:

- Accurate Motion Forecasting: Even in the face of real-world complexities, the system accurately predicts future motions of both the robot arm and surrounding objects. This is key for enabling high-level decision-making and adaptive action execution.

- Real-World Generalization: The model demonstrates strong generalization capabilities, transferring learned predictions from the benchmark environment to practical settings without requiring extensive retraining. The system's robustness in handling real-world dynamics proves the versatility of the proposed architecture.

By leveraging these visualizations and predictions in both the simulated and real environments, we show that our framework can bridge the gap between theoretical modeling and real-world robotic applications. This provides a powerful tool for task generalization, enabling robots to efficiently

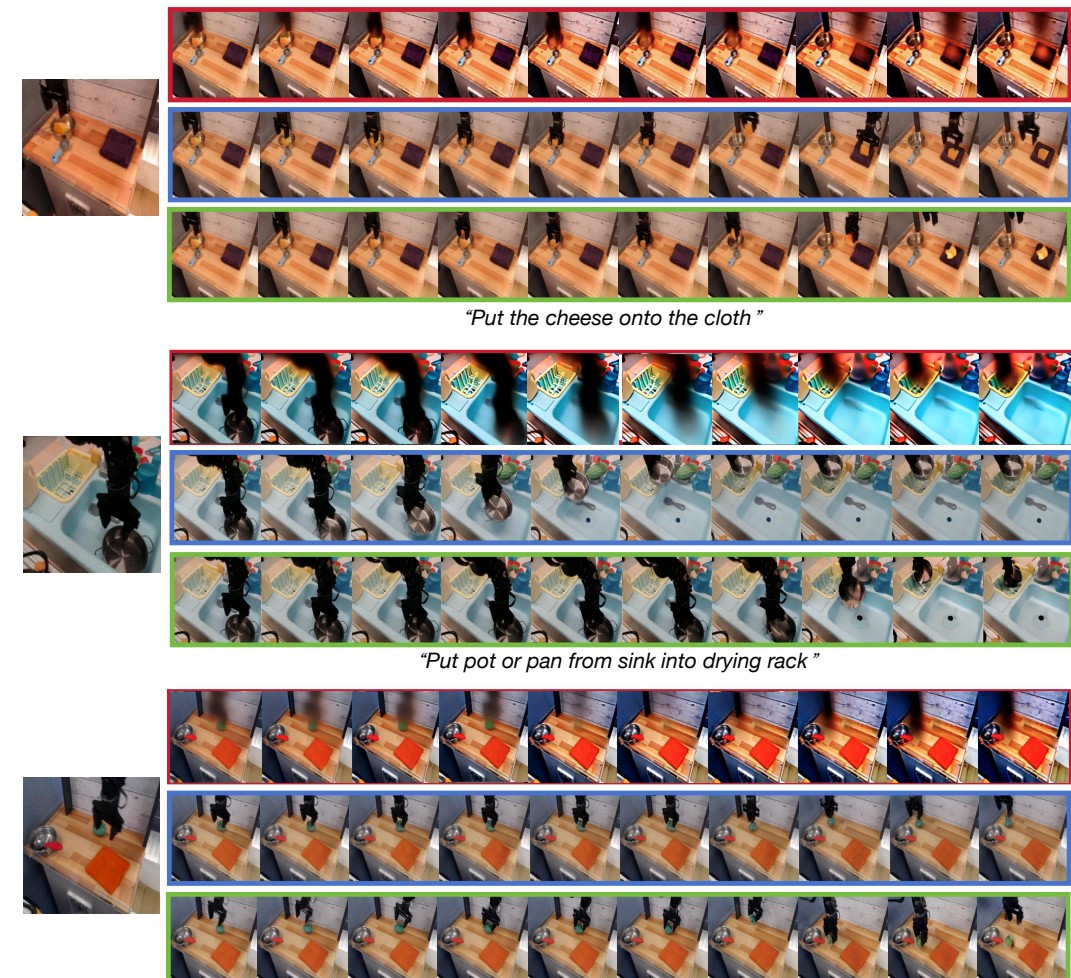

*"Put the cheese onto the cloth"*

*"Put pot or pan from sink into drying rack"*

*"Put the broccoli next to the bowl"*

Figure 7: **Visualization of Episodic Dynamics Perception on the Bridge Benchmark.** The red box indicates one-step prediction, the blue box corresponds to full-step prediction, and the green box marks the ground truth. We can observe that the representation can provide valuable information on physical dynamics, although the textures and details are not precise enough.

plan and execute complex actions in diverse scenarios. In summary, the combination of simulated and real-world results not only validates the robustness of our prediction framework but also underscores its potential for real-time action learning and autonomous decision-making in physical environments. The visualizations of future predictions further support the importance of incorporating dynamic modeling into robotic systems, fostering a deeper understanding of physical interactions and improving the overall system performance.

# F    QUALITATIVE ANALYSIS AND RESULTS

We provide qualitative examples of action sequences generated by TriVLA in Figure 10. Given multiple consecutive instructions—for instance, "Pull the handle to open the drawer," "Grasp and lift the pink block," "Use the switch to turn on the light bulb," and "Store the grasped block in the sliding cabin"—TriVLA demonstrates the ability to comprehend instructions, infer intent, and utilize predictive capabilities to accomplish long-horizon tasks. The results demonstrate that TriVLA employs VLMs and VDMs for both high-level reasoning based on common knowledge and dynamic predictive representation provided by a world model. TriVLA integrates world knowledge to enhance intent understanding and utilizes a world model for future state prediction when processing multiple sequential instructions, thereby enabling effective long horizon task execution.

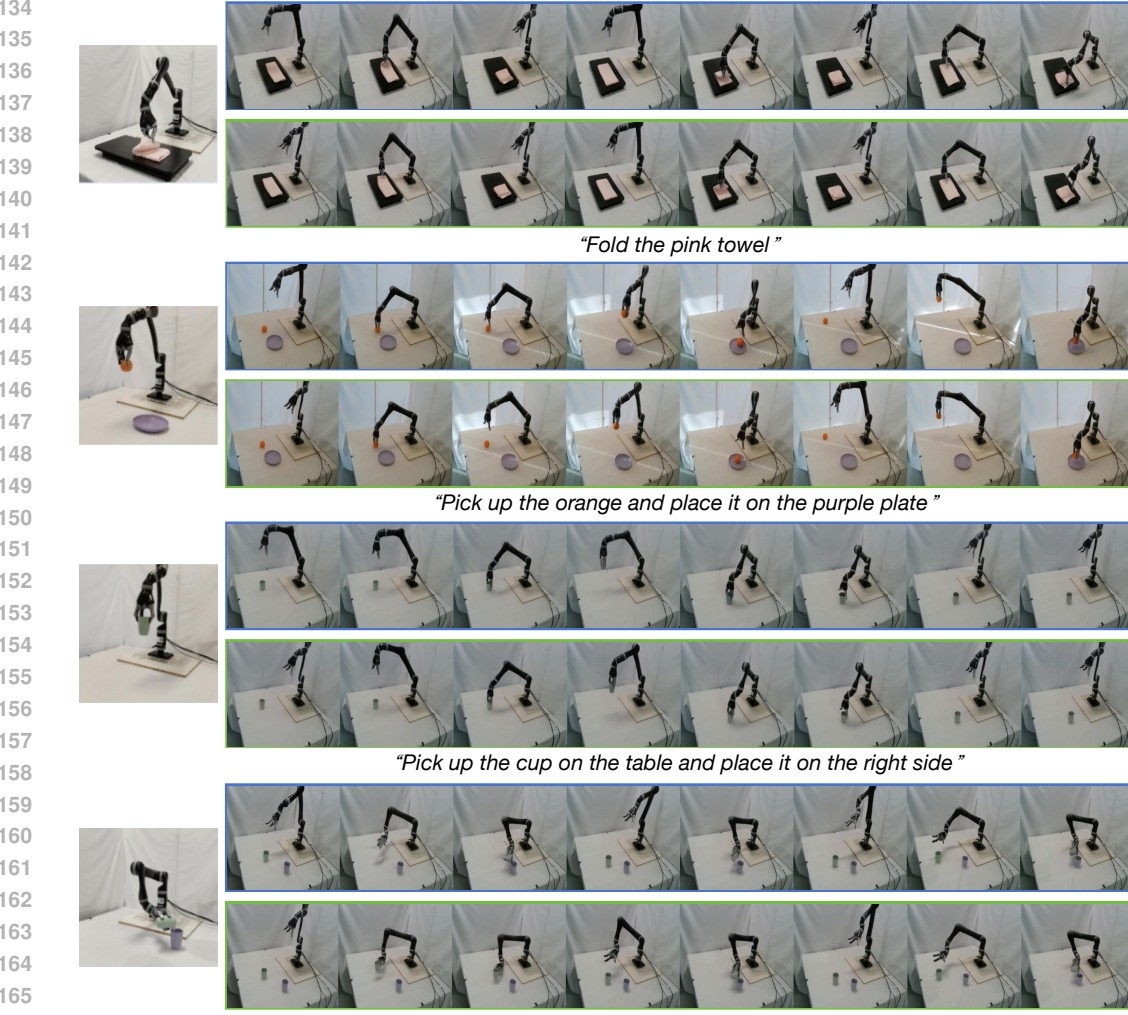

*"Fold the pink towel"*

*"Pick up the orange and place it on the purple plate"*

*"Pick up the cup on the table and place it on the right side"*

*"Pick up the green cup and pour the water into the purple cup"*

Figure 8: **Visualization of Episodic Dynamics Perception on Real-world Tasks.** The blue box corresponds to the full-step prediction, and the green box marks the ground truth of the current timestep. We can observe that representation can provide valuable information on physical dynamics, although the textures and details are not precise enough.

- Precise Alignment with Modeled Dynamics: Since simulation provides deterministic or near-deterministic dynamics, TriVLA demonstrates highly consistent outcomes across repeated trials. For instance, during multi-step manipulation sequences, object trajectories match predicted states almost perfectly, showcasing the model's capability to exploit stable environments for accurate planning.

- Stress Testing under Controlled Perturbations: Simulation allows for the systematic injection of domain variations, such as randomized object masses, altered friction coefficients, or unexpected collisions. TriVLA adapts to these controlled perturbations by updating its predictions accordingly, highlighting its resilience under a wide spectrum of simulated uncertainties.

- Long-Horizon Reasoning at Scale: Most importantly, simulation provides an ideal platform for TriVLA to validate long-horizon planning strategies across diverse scenarios. By anticipating future states over extended horizons, the model learns to optimize its policy in a wide variety of contexts, generating transferable skills that can later be fine-tuned for real-world execution.

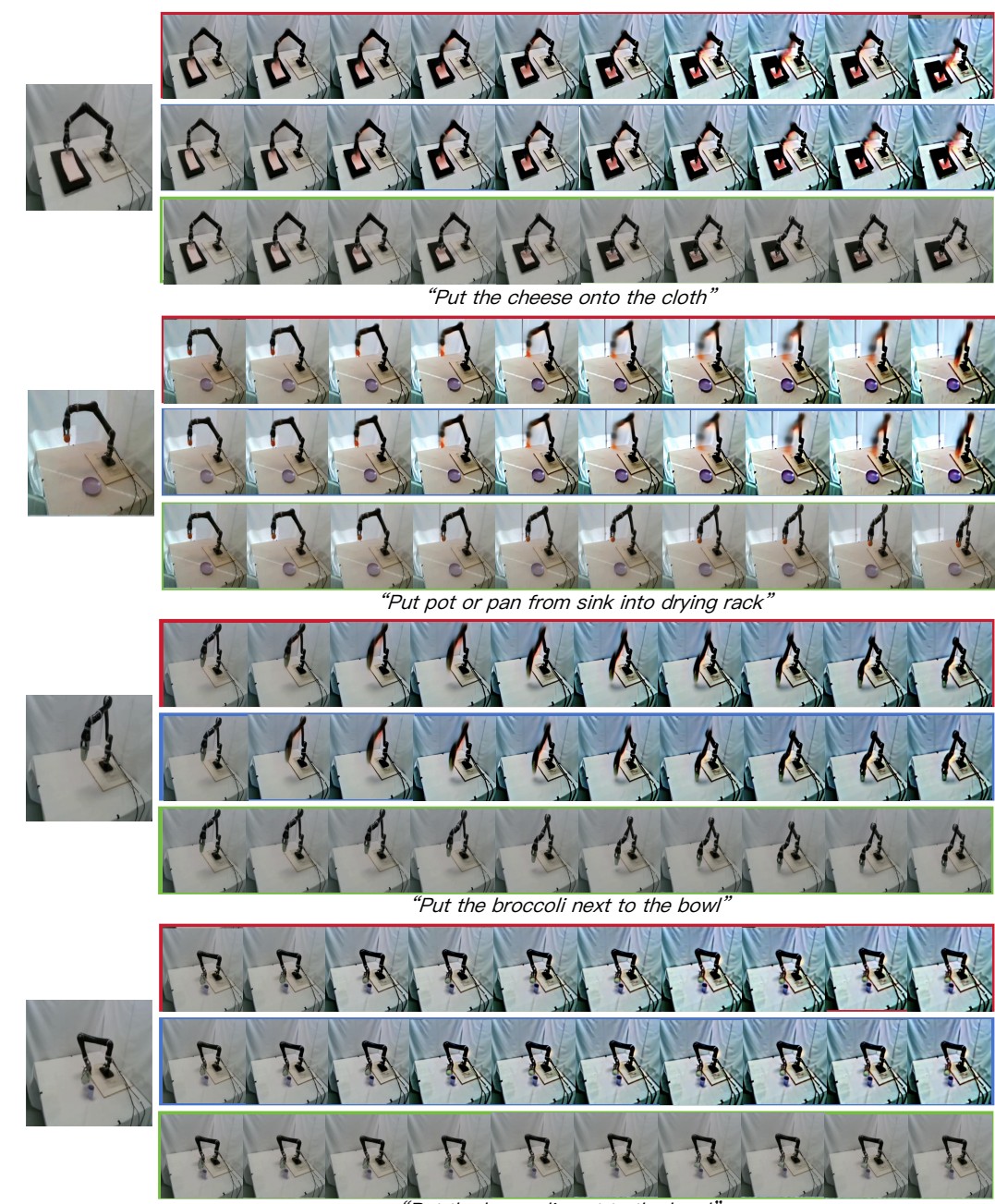

Figure 9: **Visualization of Episodic Dynamics Perception on Real-world Tasks.** The red box indicates the one-step prediction, the blue box corresponds to the full-step prediction, and the green box marks the ground truth. We can observe that representation can provide valuable information on physical dynamics, although the textures and details are not precise enough.

## G    REAL-WORLD EXPERIMENTS

In addition to the simulated results, we also conducted real-world experiments, where TriVLA successfully generates and executes action sequences on a physical robot. As shown in Figure 11, the robot accurately follows the same sequence of tasks, starting from pulling the handle to opening the drawer, grasping and lifting the pink block, using the switch to turn on the light bulb, and finally storing the block in the sliding cabin. These real-world results closely align with the predictions

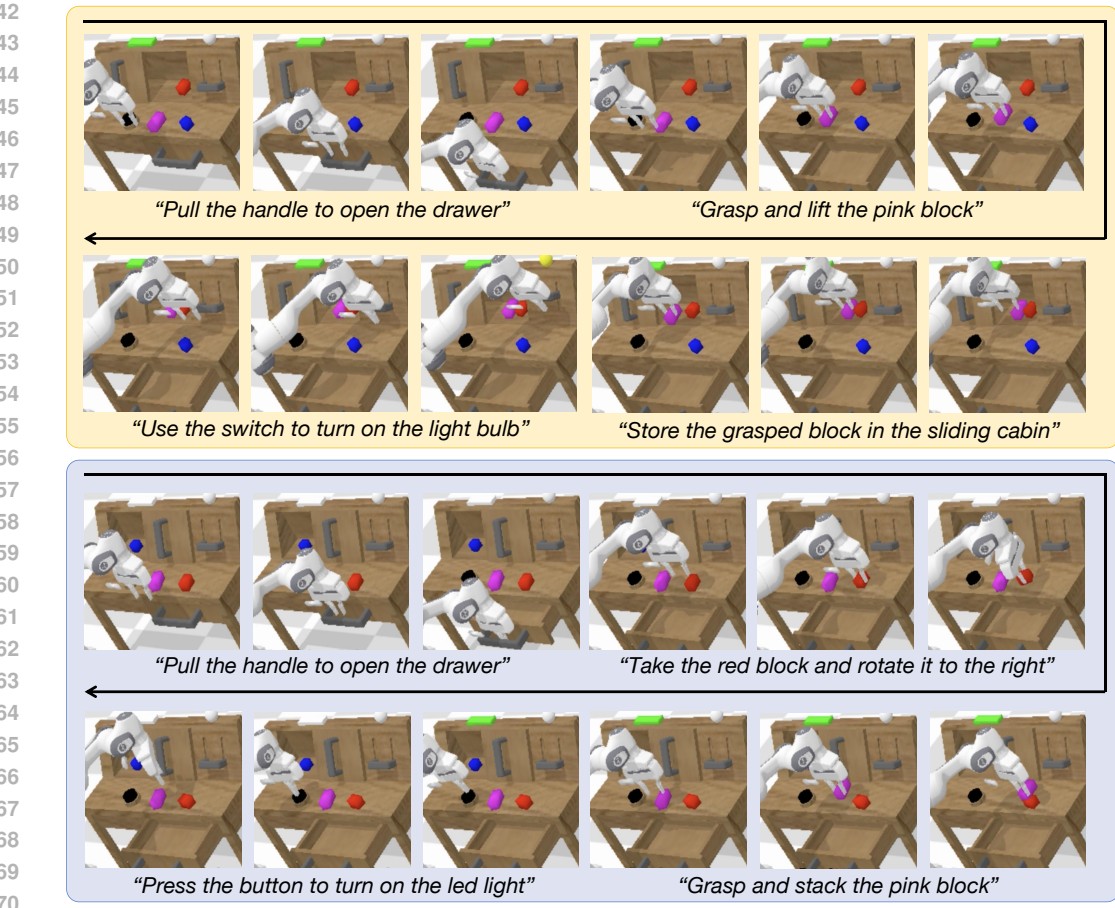

"Pull the handle to open the drawer"      "Grasp and lift the pink block"

"Use the switch to turn on the light bulb"      "Store the grasped block in the sliding cabin"

"Pull the handle to open the drawer"      "Take the red block and rotate it to the right"

"Press the button to turn on the led light"      "Grasp and stack the pink block"

Figure 10: **Qualitative case study of CALVIN benchmark.** Our **TriVLA** performs strongly in long-horizon missions. For example, in the CALVIN simulation task, it integrates world knowledge to interpret intent and uses a world model to predict future states. Given multiple sequential instructions, TriVLA can effectively execute long-horizon tasks.

and actions generated in the simulated environment, further validating TriVLA's ability to handle complex, multi-step tasks in dynamic, real-world scenarios.

The real-world experiments highlight several key advantages of TriVLA's approach:

- Real-Time Sequence Execution: The robot efficiently processes and executes long-horizon tasks in real-time, leveraging TriVLA's ability to predict intermediate states and adjust actions accordingly. Despite the inherent variability and unpredictability of the real world, such as slight environmental changes or sensor noise, TriVLA's predictive capabilities allow the robot to remain on task without requiring extensive retraining.

- High Fidelity in Task Completion: As the robot progresses through the sequence of actions, it demonstrates a strong alignment between predicted outcomes and actual results. For instance, after pulling the handle, the drawer opens correctly, and the robot adjusts its grip on the block while maintaining stability during the lift. This showcases the robustness of TriVLA's predictive world model in real-world settings.

- Dynamic Adaptation to Uncertainty: The real-world setup also presents challenges like minor inaccuracies in motor control or shifting environmental conditions. TriVLA exhibits impressive adaptability, dynamically adjusting predictions and actions to account for these uncertainties, ensuring continued task success.

- Long-Horizon Task Planning: Perhaps most notably, TriVLA demonstrates its ability to execute long-horizon plans by integrating both episodic memory and predictive reasoning. By leveraging its world model, TriVLA is able to anticipate future states and proactively

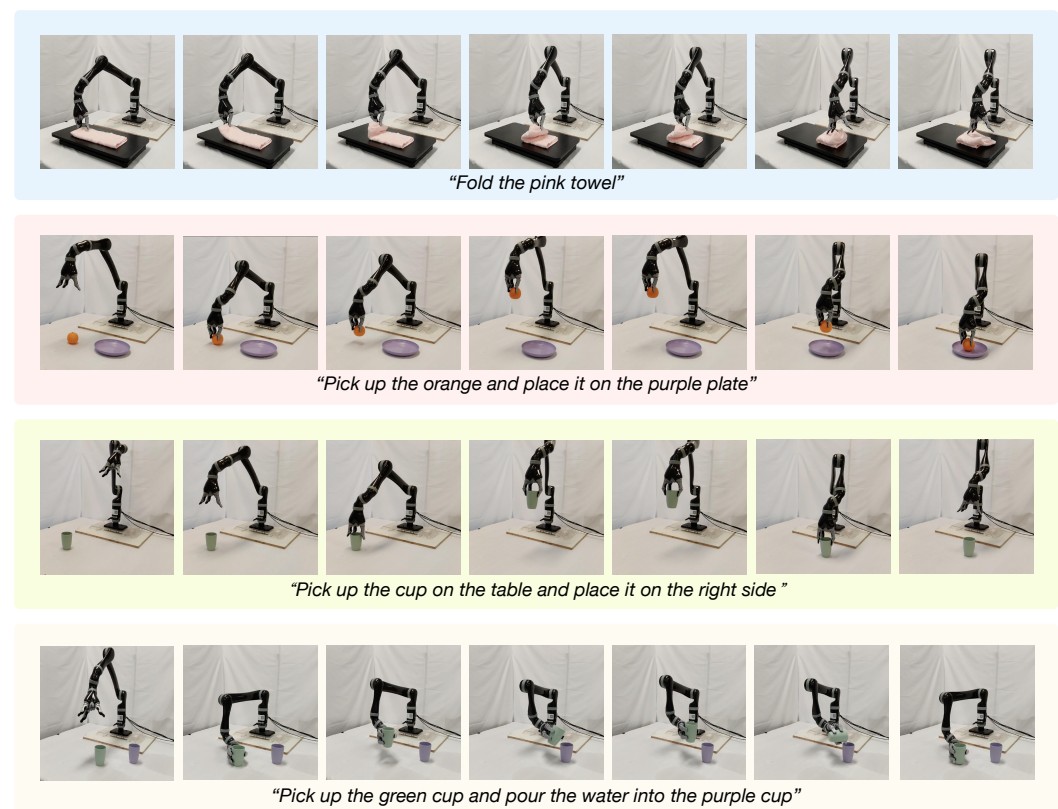

*"Fold the pink towel"*

*"Pick up the orange and place it on the purple plate"*

*"Pick up the cup on the table and place it on the right side"*

*"Pick up the green cup and pour the water into the purple cup"*

Figure 11: **Qualitative case study of real-world tasks.** Our **TriVLA** performs well in real-world tasks, successfully executing both short-horizon and long-horizon manipulations. The results illustrate its ability to integrate perception, prediction, and control for reliable task completion under real-world conditions.

> adjust actions, ensuring that all steps of the sequence are successfully completed, even in the presence of unforeseen challenges.

Overall, these real-world experiments reinforce the core strength of TriVLA: its ability to understand complex instructions, reason about sequential actions, and predict future states—an essential capability for enabling embodied agents to perform sophisticated tasks autonomously and effectively in the real world. Through the combination of simulated and real-world action sequence generation, TriVLA proves to be a highly capable architecture for long-horizon task execution, paving the way for more advanced and adaptable autonomous systems.

