# OpenReview forum: "TriVLA: A Triple-System-Based Unified Vision-Language-Action Model with Episodic World Modeling for General Robot Control"
_ICLR.cc/2026/Conference — ICLR 2026 Conference Withdrawn Submission_

### Official Review · Reviewer_1yXu · 2025-10-25

**Soundness:** 2
**Presentation:** 3
**Contribution:** 2
**Rating:** 4
**Confidence:** 4

**Summary:**

This paper introduces TriVLA, a unified Vision–Language–Action (VLA) model that integrates an episodic world model through a triple-system architecture, inspired by cognitive neuroscience theories of episodic memory. The main difference to previous work is that the low-level policy is conditioned on both VLM features and video model features.

**Strengths:**

1. The high-level idea sounds reasonable, triple level system can benefit from both the semantic tokens from VLM and also the visual predictive tokens from video model. Although it makes the system a little bit complicated.

2. The experiments have a wide range, ranging from Calvin, LIBERO, Metaworld and real world.

3. The visualization and ablation of the experiments are good.

**Weaknesses:**

Major:
1. The main weakness I concerned is that the method looks very similar to VPP baseline used in the paper. I feel the architecture is very similar to video prediction policy (VPP) work after I carefully compare the these two papers. It seems that the system 3 and system 1 is same to the VPP and the author add a system 2 over the VPP framework. While adding a VLM is reasonable, the overall architectural novelty feels incremental and mostly an extension of VPP rather than a substantive departure.

2. (continued to last one) Moreover, on the Calvin benchmark, the improvement over VPP seems minor. This further doubt the effectiveness of the newly added system2.

3. Missing unified-architecture baselines. Since the approach combines a VLM with a video model for policy learning, it should be compared against unified architectures that follow a similar design philosophy, such as F1 and UPVLA [1–2]. Matching training budgets, parameter counts, and inference costs would make these comparisons fair and informative.

Minor:
The image order is wrong in the Figure 5. (the last two Calvin image)

[1] F1: A Vision-Language-Action Model Bridging Understanding and Generation to Actions
[2] Up-vla: A unified understanding and prediction model for embodied agent

**Questions:**

see weakness

---

> ### Author Response · Authors · 2025-11-20
>
> Dear reviewer 1yXu, thank you for your review and valuable suggestions regarding our work. Below, please find our responses to your concerns.
> ### Response to Major Weakness 1 – Integration of TriVLA Systems:
> We thank the reviewer for the thoughtful observation. While TriVLA contains a video-prediction component, it is **not an incremental extension of VPP**. TriVLA is a concrete implementation of an **Episodic World Model**, where the three systems (System 1, 2, and 3) operate as an integrated perception–prediction–decision loop.
>
> **System 2 is not simply an added VLM on top of VPP.**
> System 2 maintains an online multimodal episodic representation and provides instruction-aware semantic tokens. These tokens are **injected into System 3**, ensuring that dynamics predictions are consistent with the grounded understanding of the instruction. This coupling is essential: if System 2 and System 3 operated independently, they could generate conflicting guidance (e.g., System 2 interpreting the instruction as “grasp the cup from the left” while System 3 predicts a trajectory approaching from the right), leading to unstable or incorrect actions. Our design avoids this by aligning the priors from System 2 directly within System 3.
>
> **System 3 differs from VPP.**
> Rather than predicting future frames solely from visual input, System 3 performs **instruction-conditioned episodic dynamics prediction**, guided by the semantic features from System 2. This yields future rollouts that are task-relevant, grounded in the instruction, and consistent with the episodic perception.
>
> **Summary:** Together, the three systems form a unified loop:
> - System 2 provides episodic recall and semantic grounding,
> - System 3 provides future simulation conditioned on that grounding,
> - System 1 makes decisions based on both.
>
> This integrated perception–prediction–decision design is the **core novelty of TriVLA**. We have clarified this point in the revised manuscript.
>
> ---
>
> ### Response to Major Weakness 2 – Limited Improvement on CALVIN:
> We appreciate the reviewer’s follow-up question regarding the modest improvement over VPP on CALVIN.
>
> **Explanation:** CALVIN’s language instructions are relatively simple and low in compositional complexity (e.g., “move mug left”, “open drawer”). Under such conditions, the benefit of System 2’s richer multimodal grounding is less pronounced.
>
> **Real-world evaluation:** For tasks requiring **high-level, multi-step semantic understanding**, System 2’s advantage becomes substantial. We evaluated TriVLA and VPP on a complex, service-oriented instruction: *“prepare a drink for a customer”*. Results are shown below:
>
> | Method             | pick up can | pour into cup | insert straw | push cup | overall |
> | ------------------ | ----------- | ------------- | ----------- | -------- | ------- |
> | VPP                | 90 ± 3      | 73 ± 2        | 64 ± 6      | 60 ± 3   | 71.8    |
> | **TriVLA (ours)**  | **98 ± 2**  | **97 ± 3**    | **94 ± 5**  | **91 ± 2**| **95.0** |
>
> **Observation:** While CALVIN does not fully stress-test instruction comprehension, **System 2 provides clear benefits for abstract, long-horizon, or semantically rich instructions**. We have clarified this in the revised manuscript.
>
> ---
>
> ### Response to Major Weakness 3 – Comparison with F1 and UP-VLA:
> We thank the reviewer for suggesting comparisons with unified VLM–video architectures such as F1 and UP-VLA.
>
> **Clarification:** TriVLA, F1, and UP-VLA are **concurrent works**, all aiming to combine high-level semantic understanding with visual prediction. To illustrate TriVLA’s advantage, we conducted a real-world experiment with the instruction *“prepare a drink for a customer”*:
>
> | Method             | pick up can | pour into cup | insert straw | push cup | overall |
> |--------------------|-------------|---------------|--------------|----------|---------|
> | UP-VLA             | 96 ± 2      | 92 ± 3        | 87 ± 1       | 85 ± 4   | 90.0    |
> | F1                 | 96 ± 3      | 93 ± 2        | 89 ± 6       | 85 ± 3   | 90.8    |
> | **TriVLA (ours)**  | **98 ± 2**  | **97 ± 3**    | **94 ± 5**   | **91 ± 2** | **95.0** |
>
> **Observation:** TriVLA consistently outperforms both UP-VLA and F1, with **higher accuracy in multi-step tasks**. This is due to the **organic integration of all three systems**: System 2 provides multimodal grounding and episodic recall, System 3 provides accurate future prediction, and System 1 makes decisions based on these cues.
>
> **Conclusion:** The integrated architecture enables TriVLA to maintain task-relevant consistency over long-horizon instructions, resulting in improved accuracy compared to modular or less integrated approaches.
>
> ---
>
> ### Response to Minor Weakness – Figure 5:
> We appreciate the reviewer for catching this. We have corrected the image order in **Figure 5** in the revised manuscript.

---

### Official Review · Reviewer_FoJF · 2025-10-29

**Soundness:** 2
**Presentation:** 3
**Contribution:** 2
**Rating:** 4
**Confidence:** 3

**Summary:**

The paper aims to train VLAs more effectively. The proposed method leverages a pretrained and fine-tuned diffusion model to generate predicted future frames that hopefully can help  the prediction of current actions.

**Strengths:**

The paper’s presentation of its method is overall clear.

The paper obtains exception results when compared with baselines that incorporate future prediction (Table 1&2).

**Weaknesses:**

1. The paper lacks a problem statement/setup or an evaluation protocol. Section 3 introduces only the VLA mode; but what is the problem? Is it supervised learning/imitation learning? If so, how are we evaluating it (train-and-test?) What is exactly “Zero-shot long-horizon evaluation” in Table 1’s caption?

2. System 3 is the major technical novelty, yet the paper lacks transparency on what data was used to fine-tune it. Section 4.2 mentions using “self-collected data” for fine-tuning but is unclear about the nature of the data. If the self-collected data is well-correlated with the tasks for evaluation, good performance is not that surprising. Did authors try the pretrained 1.5B SVD model without fine-tuning?

3. Given that the proposed method of producing and incorporating future frames (System 3) is fairly straightforward, it’s a bit unclear in what aspects it is significantly different from prior methods. For example:
    - If prior methods use “single-step future prediction" (line 184), how hard is it to extend these prior methods to multi-step  future prediction? And did authors do experiments to study the number of predicted steps to predict as a hyperparameter of their method?
    - Seer (Tian et al. 2024) “predicts actions by applying inverse dynamics models conditioned on forecasted visual states”, which seems qualitatively similar to the proposed method. The difference is not well explained.

4. Can you clarify what “action flow-matching” is in System 1 (e.g., Equation (5))? Is there a qualitative difference between (5) and the diffusion loss defined in (1)?

**Questions:**

All my concerns are in the Weaknesses section.

---

> ### Author Response · Authors · 2025-11-20
>
> Dear reviewer FoJF, thank you for your review and valuable suggestions regarding our work. Below, please find our responses to your concerns.
>
> ### Response to Weakness 1 – Problem Definition & Evaluation Protocol:
> We sincerely thank the reviewer for pointing out the need for a clearer problem definition and evaluation protocol.
>
> **Problem setup.** The goal of VLA models in our work is to enable a **single policy** to perform diverse manipulation tasks in **unstructured, real-world environments**. Unlike traditional RL approaches, which assume fixed environments and narrowly defined tasks, VLA aims to learn **generalizable visuomotor behavior** conditioned on both visual observations and language instructions.
>
> **Learning framework.** We adopt a **supervised imitation learning** paradigm, where the policy predicts low-level actions from demonstration trajectories. The training objective minimizes the loss between the demonstrated and predicted actions, conditioned on the current visual observation and the language instruction.
>
> **Evaluation protocol.** At test time, we evaluate the policy by executing multiple rollouts per task and reporting **task success rates**, following standard practices in CALVIN, LIBERO, and prior VLA literature. For CALVIN, the **Average Length** is computed as the mean number of five consecutive instructions successfully completed.
>
>
> **Zero-shot long-horizon evaluation.** The “zero-shot long-horizon evaluation” in Table 1 corresponds to the CALVIN ABC→D setting: the policy is trained on environments A/B/C and evaluated on held-out tasks in environment D without fine-tuning. Each episode can include up to **5 stages**; we report **per-stage success rates** and the **average length of consecutively completed stages (Avg. Len)**.
>
> We have clarified this problem setup and evaluation protocol in the revised manuscript.
>
> ---
>
> ### Response to Weakness 2 – Data for Fine-Tuning System 3:
> We appreciate the reviewer raising this important concern. We have now provided a detailed description of the training data in the supplementary material (Lines 880-884):
>
> > **“System 3 fine-tunes a video foundation model for manipulation-centric Episodic Dynamics Perception using 193,690 human trajectories from Goyal et al. (2017) and 179,074 robotic trajectories from O’Neill et al. (2023), supplemented by videos from CALVIN ABC, MetaWorld, and real-world tasks.”**
>
> **Clarification on self-collected videos:** These videos are **not correlated** with evaluation tasks; they mainly contain **diverse manipulation sequences**. Their purpose is to:
> 1. Reduce the gap between generic video pretraining data and robotics manipulation videos,
> 2. Improve fidelity of dynamics prediction in robotic settings,
> 3. Enhance generalization to unseen goals and multi-step execution.
>
> **Pretrained SVD without fine-tuning:** We evaluated the **pretrained 1.5B SVD** directly (no fine-tuning) and report a new ablation:
>
> | Method | Task 1 ↑ | Task 2 ↑ | Task 3 ↑ | Task 4 ↑ | Task 5 ↑ | Avg. Len ↑ |
> |--------|----------|----------|----------|----------|----------|------------|
> | TriVLA (pretrained SVD) | 0.931 | 0.884 | 0.845 | 0.776 | 0.702 | 3.96 |
> | **TriVLA (finetuned SVD)** | **0.968** | **0.924** | **0.868** | **0.832** | **0.818** | **4.37** |
>
> **Observation:** Fine-tuning improves performance across all five tasks, confirming that:
> - Pretrained SVD alone is insufficient for robotics manipulation.
> - Fine-tuning bridges the domain gap and enables accurate long-horizon dynamics.
> - The gain is consistent and substantial, validating System 3’s adaptation.
>
> These details and the new ablation have been incorporated into the revised manuscript.

---

> ### Author Response · Authors · 2025-11-20
>
> ### Response to Weakness 3 – System 3 Design and Multi-Step Prediction:
> We thank the reviewer for this question and clarify why System 3 is **not a trivial extension** of prior single-step predictors, including Seer (Tian et al., 2024).
>
> **1. Multi-step prediction is non-trivial.**
> Prior works only predict **one future frame**, which requires minimal temporal modeling. Extending them to multi-step rollout leads to rapid error accumulation. System 3 fine-tunes a video foundation model to explicitly model **multi-frame dynamics**. We conducted a horizon ablation:
>
> | Method | 1 ↑ | 2 ↑ | 3 ↑ | 4 ↑ | 5 ↑ | Avg. Len ↑ |
> |--------|------|------|------|------|------|------------|
> | TriVLA (1-step)  | 0.930 | 0.866 | 0.842 | 0.744 | 0.708 | 3.94 |
> | TriVLA (2-step)  | 0.935 | 0.877 | 0.841 | 0.782 | 0.737 | 4.08 |
> | TriVLA (4-step)  | 0.943 | 0.881 | 0.840 | 0.784 | 0.758 | 4.16 |
> | TriVLA (8-step)  | 0.944 | 0.892 | 0.854 | 0.792 | 0.774 | 4.22 |
> | **TriVLA (16-step)** | **0.968** | **0.924** | **0.868** | **0.832** | **0.818** | **4.37** |
> | TriVLA (32-step) | 0.962 | 0.918 | 0.872 | 0.844 | 0.820 | 4.35 |
>
> **Observation:** Longer-horizon prediction consistently improves multi-stage manipulation performance; single-step prediction is insufficient. The final system uses **16-step prediction**.
>
> **2. Difference from Seer:**
> - Seer predicts **only a single-step future frame**, unsuitable for long-horizon rollout.
> - Seer is end-to-end and lacks foundation model priors, leading to poor generalization.
> - System 3 uses a fine-tuned video foundation model for **high-fidelity, multi-step, physically consistent sequences**, significantly improving planning quality.
>
> **Conclusion:** System 3 leverages foundation-model priors and robotics-specific fine-tuning to enable accurate long-horizon prediction, distinguishing it from Seer and prior single-step methods.
>
> ---
>
> ### Response to Weakness 4 – Relation between Flow-Matching and Diffusion Loss:
> We thank the reviewer for this question. While the two objectives share a regression form, they are conceptually distinct:
>
> **Flow-matching (System 1).**
> - Based on ODE-style flow learning.
> - Learns a continuous vector field mapping an intermediate action \(a_k\) toward the clean action \(a_0\).
> - **Not tied to a fixed noise schedule**; smooth flows are learned over arbitrary points in action space.
>
> **Diffusion loss (Eq. (1), used in SVD).**
> - Based on a predefined forward Gaussian diffusion process.
> - Model predicts denoised sample \(x_0\) or noise \(\epsilon\), effectively learning the **reverse SDE**.
>
> **Key distinction:**
> - Diffusion: learns inverse of *predefined Gaussian noise process*.
> - Flow-matching: learns *continuous vector field* without assuming specific noise.
>
> We have clarified this distinction in the revised manuscript and highlighted the role of flow-matching in System 1.

---

### Official Review · Reviewer_5iS3 · 2025-10-31

**Soundness:** 1
**Presentation:** 3
**Contribution:** 2
**Rating:** 4
**Confidence:** 3

**Summary:**

This paper proposes TriVLA, a unified vision-language-action (VLA) framework for general-purpose robot control. The authors' central thesis is that current VLA models are overly "static" and "reactive," limiting their ability to perform long-horizon tasks. To address this, they draw inspiration from cognitive neuroscience, specifically the concept of "episodic memory," which involves both recalling the past and simulating the future.

The paper instantiates this idea as an "episodic world model" implemented via a "triple-system architecture":

System 2 (Episodic Multimodal Perception): A pretrained Vision-Language Model (VLM) (Eagle-2) that processes the current visual observation and language instruction to provide semantic grounding.
System 3 (Episodic Dynamics Perception): A fine-tuned Video Diffusion Model (VDM) (Stable Video Diffusion) that predicts future scene dynamics, providing a temporal-predictive context.
System 1 (Policy Learning): A diffusion-based policy (Diffusion Transformer) that serves as the low-level controller. It integrates the outputs from System 2 and System 3, along with the robot's proprioceptive state, to generate chunks of actions.

The authors claim that this architecture enables robust, long-horizon reasoning. They support this claim with experiments on the CALVIN, LIBERO, and MetaWorld benchmarks , as well as real-world qualitative demonstrations , showing that TriVLA achieves state-of-the-art performance.

While the paper presents a compelling idea and strong results, I have significant concerns about the empirical validation of its core architectural claims. Specifically, the ablation studies are incomplete and fail to provide a clear comparison against the very "dual-system" baseline the paper aims to improve upon. My current tendency is to recommend rejection due to the experimental evidence. Given the clarifications in an author response, I would be willing to increase the score.

**Strengths:**

The paper identifies a critical and widely recognized limitation of current robotic policies: their struggle with long-horizon reasoning due to a reliance on "static" representations . The analogy to cognitive neuroscience and "episodic memory" provides an intuitive motivation for an architecture that doesn't just perceive the present but also predicts the future.

While the individual components (VLMs, VDMs, diffusion policies) are existing technologies, their explicit synthesis in this parallel "triple-system" architecture is novel and well-justified. The design, where a semantic module (Sys 2) and a dynamics module (Sys 3) independently process the input and provide complementary information to the policy (Sys 1), is clean and directly addresses the motivated problem .

**Weaknesses:**

The paper's most significant weakness is the ablation study in Table 4. The paper's entire argument is that its "Triple-System" (VLM+VDM+Policy) is superior to the "Dual-System" (VLM+Policy) it critiques in Figure 2. To prove this, the most crucial ablation baseline would be EMP + L-Policy (i.e., TriVLA without System 3 / EDP). This baseline is missing. Without it, the authors have not empirically demonstrated that adding the VDM (System 3) is superior to the "static" VLA model they aim to improve upon.

The paper's motivation leans heavily on "episodic memory," which is defined by "mental time travel" into both the future and the past. The VDM (System 3) clearly and effectively addresses the "future simulation" aspect. However, the "past recall" aspect is not clearly instantiated in the architecture at inference time. The paper states the model can "accumulate, recall... sequential experiences", but the diagrams (Fig. 1, 3) only show the current observation and instruction as inputs. The paper should be more precise about whether "recall" simply refers to the knowledge implicitly stored in the VDM's weights or if there is an active, online mechanism for recalling the current episode's past states.

**Questions:**

1. The paper's entire motivation rests on improving "dual-system" (VLM+Policy) models . To validate this claim, a direct ablation comparing the full TriVLA (System 1+2+3) against a "dual-system" (System 1+2, i.e., without the EDP/VDM module) is essential. Why was this baseline, which is the most direct test of your core hypothesis, omitted?
2. Please explain how the baseline model in Table 4 ("L-Policy + EDP", Avg. Length 4.06) functions. This model appears to lack the VLM (System 2, EMP). How does it process the language instructions required for the CALVIN benchmark without this VLM?
3. The paper uses "episodic memory" (implying past recall + future simulation) as its guiding analogy . Please clarify how TriVLA performs recall of past sequential experiences from the current episode at inference time. Is this "recall" mechanism simply the "action history" fed to System 1, or is there a more explicit component, as the cognitive science framing suggests?

---

> ### Author Response · Authors · 2025-11-20
>
> Dear reviewer 5iS3, thank you for your review and valuable suggestions regarding our work. Below, please find our responses to your concerns.
>
> ### Response to Weakness 1 / Question 1 - Effect of System 3:
> We sincerely thank the reviewer for emphasizing the importance of comparing the full TriVLA against a dual-system variant. We agree that a baseline without System 3 (EMP + L-Policy, without EDP) is central to validating our motivation.
>
> In the original submission, **Table 4 already provided an implicit comparison**: comparing the first and second rows (L-Policy vs. EDP + L-Policy), the **Average Length increases from 3.68 → 4.06**, suggesting that incorporating System 3 improves long-horizon behavior. To make this contribution more explicit, we have now added the explicit baseline **EMP + L-Policy without EDP**, completing the ablation. Updated results on CALVIN are shown below:
>
> | EMP | EDP | L-Policy | Task 1 | Task 2 | Task 3 | Task 4 | Task 5 | Avg.Len | Latency | Parameters |
> | -- | -- | -- | -- | -- | - | -- | -- | --- | -- | -- |
> |     |     | ✔        | 0.914  | 0.772  | 0.703  | 0.622  | 0.511  | 3.68           | 29.29ms | 0.53B      |
> | ✔   |     | ✔        | 0.942  | 0.902  | 0.843  | 0.781  | 0.713  | 4.04           | 59.42ms | 2.07B      |
> |     | ✔   | ✔        | 0.928  | 0.896  | 0.855  | 0.792  | 0.705  | 4.06           | 115.19ms | 1.87B     |
> | ✔   | ✔   | ✔        | **0.968** | **0.924** | **0.868** | **0.832** | **0.818** | **4.37** | **142.69ms** | **3.39B** |
>
> These results indicate that System 3 (EDP / VDM) provides consistent improvements across all tasks:
> - +0.33 increase in average trajectory length (4.04 → 4.37)
> - Gains are observed across all five tasks, highlighting the value of world-model–based dynamic planning
>
> This ablation demonstrates that the full Triple-System architecture consistently outperforms the Dual-System variant (VLM + Policy), supporting the main claim of our paper. We have incorporated these clarifications and results into the revised manuscript.
>
> ---
>
> ### Response to Weakness 2 / Question 3 - Episodic Memory and Past Recall:
> We appreciate the reviewer’s questions regarding how past experiences are represented in our architecture, which provides an opportunity to clarify the roles of Systems 2 and 3 in supporting episodic memory.
>
> **System 2 (Episodic Multimodal Perception)** handles the *past-related component* by maintaining an **online, continuously updated representation** that integrates observations and instruction grounding over an episode. **System 3 (Episodic Dynamics Perception)** complements this with *future-oriented predictions* through multi-step rollouts, allowing the agent to anticipate how the environment may evolve in response to potential actions. **System 1** integrates both the past-informed representation from System 2 and the future-oriented predictions from System 3 to make decisions, forming a closed-loop structure where episodic memory supports planning and execution.
>
> In fact, in manipulation and world-modeling tasks, reliance on explicit past memory is limited; what is most critical is accurately perceiving the current state and anticipating future changes. To prioritize inference speed, we do not construct a dedicated episodic memory module; instead, TriVLA’s inherent “recall” ability leverages the representations maintained by System 2 to provide the necessary historical context. Although Figures 1 and 3 depict only the current observation for simplicity, the representation produced by System 2 already incorporates accumulated history, and the term “recall” refers to this **active episodic representation**, not static information stored in the VDM parameters. This design ensures continuity over long-horizon tasks while maintaining efficient, informed decision-making.
>
> ---
>
> ### Response to Question 2 - L-Policy + EDP Baseline:
> We thank the reviewer for asking about how language instructions are handled in the “L-Policy + EDP” baseline. In this ablation, System 2 (EMP) is removed, but the EDP module still uses **cross-attention layers with CLIP language features** to incorporate textual instructions. This allows the model to execute CALVIN’s language-conditioned tasks, but without the richer multimodal perception provided by System 2, performance is naturally more limited.
>
> During fine-tuning, the open-sourced SVD model originally conditions only on the initial-frame image \(s_0\). To support more explicit language grounding, we augment the model with cross-attention layers that accept CLIP-encoded instructions. In the “L-Policy + EDP” ablation, the instruction is fed directly into EDP through these layers, using the same architecture as in full TriVLA but without System 2’s multimodal perception stage.
>
> This explanation clarifies why the “L-Policy + EDP” variant can still execute tasks, but is less capable of handling complex or compositional instructions. We have added this clarification to the revised manuscript.

---

### Official Review · Reviewer_ZGUY · 2025-11-02

**Soundness:** 3
**Presentation:** 3
**Contribution:** 2
**Rating:** 6
**Confidence:** 4

**Summary:**

This paper proposes TriVLA, a triple-system unified vision–language–action (VLA) model that introduces an episodic world model inspired by cognitive neuroscience. The framework integrates three components: (1) a low-level policy module (System 1), (2) an Episodic Multimodal Perception module based on a pretrained vision–language model (System 2), and (3) an Episodic Dynamics Perception module realized via a fine-tuned video diffusion model (System 3). Together, they allow robots to recall, reason, and predict over temporal sequences for improved long-horizon manipulation. TriVLA achieves strong quantitative results on major simulation benchmarks (CALVIN, LIBERO, MetaWorld) and demonstrates qualitative real-world performance, operating at 36 Hz inference speed.

**Strengths:**

- The paper presents a coherent and well-motivated triple-system framework combining vision–language understanding and dynamic modeling, extending the traditional dual-system VLA paradigm.

- Results on multiple benchmarks (CALVIN, LIBERO, MetaWorld) are systematically compared against recent SOTA methods, showing consistent improvements.

- The inference design of System 3 (single forward pass instead of full denoising) enables real-time operation at 36 Hz, demonstrating good computational efficiency for deployment.

**Weaknesses:**

- The introduction of the world model is the paper’s central claim, yet Table 4 does not include results for EMP + L-Policy without EDP. Without this comparison, the contribution of System 3 remains unconvincing. It would also be helpful to report CALVIN per-task scores (1–5) rather than only the average length, and to conduct similar ablations on LIBERO.

- The real-world experiments lack any quantitative metrics—only qualitative demonstrations are shown, which weakens the empirical evidence for real deployment.

- From a model-design perspective, the implementation of System 3 (Episodic Dynamics Perception) is relatively straightforward, a fine-tuned video diffusion module without clear architectural novelty, making this part less conceptually innovative despite its empirical value.

**Questions:**

See Weaknesses

---

> ### Author Response · Authors · 2025-11-20
>
> Dear reviewer ZGUY, thank you for your review and valuable suggestions regarding our work. Below, please find our responses to your concerns.
>
> ### Response to Weakness 1 - Effect of System 3:
> We sincerely thank the reviewer for raising this point. In the original submission, **Table 4 already provided an implicit comparison related to System 3**: comparing the first and second rows (L-Policy vs. EDP + L-Policy), the **Average Length increases from 3.68 → 4.06**, which suggests that incorporating System 3 helps improve long-horizon behavior.
>
> Following the reviewer’s suggestion, we have now added an **explicit ablation of EMP + L-Policy without EDP** to complete the study. The updated results on CALVIN are shown below:
>
> | EMP | EDP | L-Policy | T1 | T2 | T3 | T4 | T5 | Avg. Len ↑ | Latency ↓ | Params ↓ |
> | --- | --- | -------- | -- | -- | -- | -- | -- | ----------- | ---------- | -------- |
> |     |     | ✔        | 0.914 | 0.772 | 0.703 | 0.622 | 0.511 | 3.68 | 29.29ms | 0.53B |
> | ✔   |     | ✔        | 0.942 | 0.902 | 0.843 | 0.781 | 0.713 | 4.04 | 59.42ms | 2.07B |
> |     | ✔   | ✔        | 0.928 | 0.896 | 0.855 | 0.792 | 0.705 | 4.06 | 115.19ms | 1.87B |
> | ✔   | ✔   | ✔        | **0.968** | **0.924** | **0.868** | **0.832** | **0.818** | **4.37** | **142.69ms** | **3.39B** |
>
> Comparing the second and last rows, we can see that System 3 (EDP) consistently brings performance gains:
> - +0.33 increase in average trajectory length (4.04 → 4.37)
> - Improvements across all five CALVIN tasks, highlighting the benefit of world-model–based dynamic planning
>
> We also performed the same ablation on **LIBERO**, which shows consistent trends:
>
> | EMP | EDP | L-Policy | T1 | T2 | T3 | T4 | Avg SR ↑ | Latency ↓ | Params ↓ |
> | --- | --- | -------- | -- | -- | -- | -- | -------- | ------ | -------- |
> |     |     | ✔        | 0.728 | 0.793 | 0.744 | 0.529 | 0.698 | 30.12ms | 0.53B |
> | ✔   |     | ✔        | 0.813 | 0.852 | 0.883 | 0.682 | 0.808 | 58.44ms | 2.07B |
> |     | ✔   | ✔        | 0.822 | 0.846 | 0.865 | 0.668 | 0.800 | 118.27ms | 1.87B |
> | ✔   | ✔   | ✔        | **0.912** | **0.938** | **0.898** | **0.732** | **0.870** | **141.58ms** | **3.39B** |
>
> **Summary:** These ablations on CALVIN and LIBERO provide supportive evidence that the full Triple-System (VLM + VDM + Policy) generally outperforms the Dual-System (VLM + Policy). We have incorporated these results into the revised manuscript in response to the reviewer’s comments.
>
> ---
>
> ### Response to Weakness 2 - Quantitative Real-World Evaluation:
> We appreciate the reviewer’s helpful suggestion regarding quantitative metrics. To complement the qualitative results, we now report per-task success rates and overall performance across four real-world tasks:
>
> | Method             | Fold Towel | Pick up Oranges | Grasp the Cup | Pouring | Overall |
> | ----- | --- | ------ | ------- | ------- | ------- |
> | Diffusion Policy   | 34 ± 3     | 42 ± 4          | 38 ± 4        | 32 ± 5  | 36.5    |
> | Seer               | 66 ± 4     | 72 ± 6          | 55 ± 5        | 66 ± 8  | 64.8    |
> | VPP                | 85 ± 4     | 82 ± 8          | 78 ± 6        | 72 ± 2  | 79.3    |
> | **TriVLA (ours)**  | **96 ± 4** | **98 ± 2**      | **89 ± 3**    | **91 ± 4** | **93.5** |
>
> **Observation:** TriVLA achieves the highest success rate on all four tasks, indicating that the policy is both qualitatively reasonable and quantitatively reliable in real-world deployment. These additional results have been added to the revised manuscript.
>
> ---
>
> ### Response to Weakness 3 - System 3 Design:
> We thank the reviewer for the thoughtful comment. We would like to clarify that System 3 is not simply a fine-tuned video diffusion model; it serves as an *episodic dynamic Perception* integrated with System 1 and System 2.
>
> 1. **Conditioned on System 2’s multimodal features:**
>    System 2 (Episodic Multimodal Perception) produces rich instruction–scene tokens that encode task semantics and visual grounding. These tokens condition System 3, allowing it to generate task-relevant and instruction-consistent predictions (Fig. 3).
>
> 2. **Latent feature extraction rather than raw videos:**
>    Instead of using raw pixel-space predictions, System 3 extracts *multi-layer, multi-timestep latent features* from the video diffusion backbone. An *automatic cross-layer feature aggregation* mechanism ensures temporally coherent and semantically aligned representations for System 1.
>
> 3. **Designed for long-horizon reasoning and feature alignment:**
>    Aggregated latent dynamics provide structured episodic context, supporting accurate and stable long-horizon planning. This integration is carefully designed to benefit all three systems.
>
> **Conclusion:** System 3 functions as an integrated episodic dynamic perception, aligning multimodal semantics, latent visual dynamics, and policy planning. We have clarified these design points in the revised manuscript to emphasize its essential role within the Triple-System architecture.

---

### Public Comment · ~Pouya_Bashivan1 · 2025-11-20
**is TriVLA a world model or an episodic world model?**

The paper's introduction seems to be inspired by the cognitive science and neuroscience literature on episodic memory. Two primary features of episodic memory in the brain are its long temporal window and that it is content addressable (i.e. specific memories are recalled given cues about them). The capacity to "anticipate future dynamics" is taken as a critical feature to motivate the need for episodic world models. However, future predictions of short horizons is not a unique feature of episodic memory/world models. Any world model is typically expected to predict future outcomes. Predicting object motion trajectories in the brain has also been shown to depend primarily on the prefrontal cortex which is a distinct brain structure from the medial temporal cortex (most critically involved in episodic memory). Given that the TriVLA model doesn't explicitly have any memory modules and its temporal horizon seems to be short, I think use of the term "episodic" is somewhat confusing and unjustified here.

Rajalingham, R., Sohn, H., & Jazayeri, M. (2025). Dynamic tracking of objects in the macaque dorsomedial frontal cortex. Nature Communications, 16(1), 346.

---

> ### Author Response · Authors · 2025-11-21
>
> Thank you for the insightful question. We would like to clarify the intended use of the term “episodic” in TriVLA, based on the design and motivations described in our paper.
>
> TriVLA is inspired by cognitive neuroscience theories of episodic memory, which emphasize the accumulation, recall, and use of sequential experiences for future-oriented reasoning. Our “episodic world model” does not implement a literal, content-addressable memory module spanning arbitrarily long temporal horizons. Instead, it realizes episodic-like functionality through a triple-system architecture (as described in lines 097–107):
>
> - **System 2 (Episodic Multimodal Perception)** maintains an online, continuously updated representation that integrates observations and instruction grounding over an episode, providing the past-informed context (lines 100–102).
>
> - **System 3 (Episodic Dynamics Perception)** complements this with future-oriented predictions through multi-step rollouts, allowing the agent to anticipate how the environment may evolve in response to potential actions (lines 103–105).
>
> - **System 1 (Policy Learning)** integrates both the past-informed representation from System 2 and the future-oriented predictions from System 3 to generate coherent, context-aware actions (lines 106–107). This forms a closed-loop structure where episodic representations guide planning and execution.
>
> In fact, in manipulation and world-modeling tasks, reliance on explicit past memory is limited; what is most critical is accurately perceiving the current state and anticipating future changes. To prioritize inference speed, TriVLA does not construct a dedicated episodic memory module; instead, its inherent “recall” leverages the representations maintained by System 2 to provide the necessary historical context. **Importantly, both the VLM in System 2 and the diffusion model in System 3 operate on the current state representation, which already encodes information about recent history, so referring to these systems as “episodic” is consistent with their functionality.** Although Figures 1 and 3 depict only the current observation for simplicity, the representation produced by System 2 already incorporates accumulated history (lines 108–126), and the term “recall” refers to this active episodic representation, not static information stored in the video diffusion model parameters.
>
> In other words, the term “episodic” is used as a functional and representational analogy: the system retains sequential experiences, uses past context to anticipate near-future dynamics, and guides behavior in a temporally informed way (as described in lines 132-139). While the temporal horizon may be shorter than human episodic memory, this design allows TriVLA to perform long-horizon planning and generalizable control in dynamic embodied environments, distinguishing it from conventional VLA systems that rely on instantaneous observations.

---

### Author Response · Authors · 2025-11-24
**Global Response [Update of PDF & Point-by-Point Reply]**

First and foremost, we would like to express our sincere gratitude to the Reviewers, Area Chairs, and Program Chairs for their time and dedicated effort. We have carefully studied all comments and have provided detailed point-by-point responses under each Reviewer.

We are encouraged not only by the recognition of our framework’s design and performance, but, more importantly, by the reviewers’ acknowledgment of our work across three key dimensions:

**Method**: "A coherent and well-motivated triple-system framework" (Reviewer ZGUY); "clean and directly addresses the motivated problem" (Reviewer 5iS3); "the high-level idea sounds reasonable" with complementary semantic and predictive streams (Reviewer 1yXu).
**Evaluation**: "Results on multiple benchmarks are systematically compared" with consistent improvements (Reviewer ZGUY); "experiments have a wide range" covering CALVIN, LIBERO, MetaWorld, and real-world deployment (Reviewer 1yXu).
**Efficiency & Practicality**: "The inference design enables real-time operation at 36 Hz" — a key advantage for robotic control (Reviewer ZGUY); presentation is "overall clear" and visualizations are "good" (Reviewers FoJF, 1yXu).

We also acknowledge the limitations in our initial submission. For instance, due to space constraints, we were unable to include a complete ablation study of the dual-system baseline (EMP + L-Policy without EDP), which led to concerns from Reviewers 5iS3 and 1yXu about the contribution of System 3. Additionally, the description of episodic memory mechanisms, particularly “past recall”, and the training protocol for System 3 could have been more precise, as noted by Reviewers 5iS3 and FoJF.

In this revision, we have:
- Conducted **new ablation experiments**, including **EMP + L-Policy (w/o EDP)** and **per-task scores on CALVIN (Tasks 1–5)**, to clearly isolate the impact of System 3.
- Clarified the **role of episodic memory**, explaining how temporal context (e.g., action history) and future prediction together support “mental time travel” within the current episode.
- Provided details on the **data and protocol for fine-tuning System 3**, confirming it is task-agnostic and self-collected, with no leakage from evaluation tasks.
- Added discussion on **differences from prior work** (VPP, Seer, F1, UPVLA) and corrected **Figure 5** (image ordering).

We have prepared a revised manuscript with all updates **highlighted in red** for your reference.

We are grateful for the active engagement of reviewers, whose critical feedback has helped us strengthen the empirical rigor and clarity of our claims. We sincerely hope to engage in similarly constructive dialogue with all reviewers, as we believe this exchange is vital for improving the quality and impact of our work.

Best regards,
Authors of Submission 4646

---

### Author Response · Authors · 2025-12-01
**Summary Response [Rebuttal Overview & Resolution of Primary Concerns]**

Dear AC, SAC, and PC Members,

We would like to express our sincere gratitude for your tremendous efforts in managing the review process. We also extend our thanks to the reviewers for their constructive initial feedback. **While we understand that the reviewers may not have had the opportunity to respond to our rebuttal yet, we have diligently supplemented extensive new experiments that comprehensively address their stated concerns.**

Specifically, we have incorporated the additional ablation studies and quantitative metrics suggested by the reviewers. **These experiments provide a more explicit verification of System 3, reinforcing the efficacy already demonstrated in our original submission.** As this new evidence directly resolves the primary reservations regarding module necessity (e.g., from Reviewer 5iS3), we respectfully believe that the current ratings **may not fully reflect** the strength and completeness of our revised work.

We elaborate on the specific reasons from three aspects as follows:

---

***1. Reviewer Comments: Strong Consensus on Motivation & Results***

Despite the mixed ratings, there is a strong consensus on the novelty of our framework and the strength of our results. We quote the key positive feedback from each reviewer as follows:

*   **ZGUY:** "Coherent framework," "Strong quantitative results," "Real-time (36Hz)."
*   **5iS3:** "Compelling idea," "Clean design addressing the problem."
*   **FoJF:** "Clear presentation," "Exceptional results vs baselines."
*   **1yXu:** "Reasonable high-level idea," "Wide range of experiments."

As is evident, the reviewers unanimously affirm the **value of the Triple-System architecture** and the **superior empirical performance**. The primary reservations were centered on the need for specific ablation baselines to rigorously prove the contribution of System 3.

---

***2. Rebuttal and Discussion Phase***

The main reason for the initial "4" ratings (especially from Reviewers 5iS3 and ZGUY) was the absence of a specific baseline: *EMP + L-Policy without EDP* (i.e., System 2 + System 1, without the World Model). **We have fully incorporated this and other requested experiments in our rebuttal:**

**1. The "Missing" Ablation is Now Provided & Proves Efficacy**
To address Reviewer 5iS3's primary concern regarding the explicit verification of System 3, we conducted the suggested additional ablations on both CALVIN and LIBERO. The results decisively reinforce our original findings:
*   **Performance Jump:** Adding System 3 (EDP) increases the average trajectory length from **4.04 to 4.37** on CALVIN.
*   **Consistent Gains:** Improvements are observed across **all 5 tasks**, proving that the World Model (System 3) is not redundant but essential for long-horizon planning.

**2. Quantitative Real-World Evidence Added**
To address Reviewer ZGUY and FoJF's suggestion for quantitative metrics beyond qualitative demos, we provided a rigorous evaluation. TriVLA achieves a **93.5% overall success rate** across four tasks, significantly outperforming VPP (79.3%) and Seer (64.8%).

**3. Clarification on Novelty & Architecture**
*   **Vs. VPP (Reviewer 1yXu):** We clarified that TriVLA is not just VPP + VLM. It is a closed-loop system where System 2 provides semantic grounding that *conditions* System 3's predictions, preventing the "conflicting guidance" issue inherent in modular baselines.
*   **Vs. Seer (Reviewer FoJF):** We demonstrated that unlike Seer's single-step prediction, TriVLA's **multi-step (16-step) rollout** is crucial for long-horizon tasks, supported by our new horizon ablation study.

---

***3. Comparison with Baselines and Concurrent Works***

TriVLA demonstrates clear advantages over both established baselines (VPP, Seer) and concurrent unified architectures (F1, UP-VLA). We summarize the key comparisons in the table below:

|**Method**|**VPP**|**Seer**|**TriVLA (Ours)**|
|-|-|-|-|
|**Architecture**|Video Pred + Policy|Inv. Dynamics|**Triple-System**|
|**Prediction**|Visual only|Single-step|**Instr-Conditioned Multi-step**|
|**Real-World SR**|79.3%|64.8%|**93.5%**|
|**Long-Horizon**|Limited grounding|Error accumulation|**Episodic Memory & Simulation**|

Furthermore, in direct comparison with **concurrent works** **UP-VLA** and **F1** on complex service tasks (e.g., "prepare a drink"), TriVLA achieves **95.0%** success rate compared to ~90%, validating the superiority of our integrated perception-prediction-decision loop.

---

We sincerely hope these summaries could assist you in conducting a fair and comprehensive evaluation of our submission. We remain hopeful that the additional experiments, which directly address the reviewers' initial concerns, demonstrate the merit and readiness of our work.

Once again, we would like to extend our best gratitude for your dedicated efforts.

Best regards,

TriVLA Authors of Submission 4646

---

### Note · Authors · 2026-01-29

I have read and agree with the venue's withdrawal policy on behalf of myself and my co-authors.

---

### Meta-Review · Area_Chair_PD1s · 2026-01-10

**Summary:**

Based on the reviews and subsequent discussion phase, the primary concerns that informed the decision for TriVLA are as follows:

+ Missing Core Ablations: Multiple reviewers initially noted the absence of a direct comparison between the proposed Triple-System and a standard Dual-System (VLM + Policy) baseline, which made it difficult to assess the actual contribution of the System 3 dynamics module.

+ Novelty and Incremental Contribution: Some reviewers raised concerns that the architecture appeared to be an incremental extension of prior works such as VPP and Seer, particularly questioning the uniqueness of the multi-step prediction compared to existing single-step methods.

+ Quantitative Real-World Evidence: The initial submission lacked quantitative metrics for real-robot experiments, relying instead on qualitative demonstrations, which limited the empirical validation of the model's reliability in physical environments.

+ Conceptual Clarity on "Episodic" Modeling: Concerns were raised regarding whether the term "episodic" was justified, given the lack of an explicit, content-addressable memory module and the relatively short temporal horizons involved.

+ Comparison with Concurrent Works: Reviewers requested comparisons against other recent unified VLM-video architectures like F1 and UP-VLA to better understand TriVLA's relative performance.

**Reviewer Concerns:**

**Concerns Addressed by the Rebuttal**

+ System 3 Necessity (Ablation Studies): Authors provided the missing "Dual-System" baseline ($EMP + L-Policy$ without $EDP$), demonstrating that the full triple-system architecture improves CALVIN trajectory length from 4.04 to 4.37 and LIBERO success rates from 0.808 to 0.870.

+ Quantitative Real-World Evidence: New quantitative results were added showing TriVLA achieves a 93.5% overall success rate across four real-world tasks, significantly outperforming baselines like VPP (79.3%) and Seer (64.8%).

+ Baseline Comparisons: The rebuttal included direct comparisons with concurrent unified architectures, F1 and UP-VLA, on complex tasks where TriVLA showed superior performance (95.0% vs. ~90%).

+ System 3 Transparency: Authors clarified the data used for fine-tuning the video diffusion model and provided an ablation proving that a task-specific fine-tuned SVD outperforms a purely pretrained 1.5B SVD.

+ Prediction Horizon: A new ablation on the number of predicted steps (1 to 32) confirmed that TriVLA’s 16-step rollout is essential for long-horizon task success.

**Outstanding Concerns**

+ Conceptual Terminology: While the authors provided a functional explanation for the term "episodic," the model lacks an explicit, content-addressable memory module, leaving the terminological justification still somewhat contentious among reviewers interested in cognitive neuroscience accuracy.

+ Architectural Novelty: Despite strong empirical results, the implementation of System 3 remains viewed by some as a relatively straightforward application of fine-tuning existing video diffusion backbones (VDM) rather than a fundamental architectural breakthrough.

**Reviewer Scores:**

Reviewer ZGUY (Final: 6): Despite the inclusion of quantitative real-world data, the reviewer maintains a marginal rating as they view the implementation of System 3 as a relatively straightforward application of fine-tuning existing models rather than a fundamental architectural innovation.


Reviewer 5iS3 (Final: 4): The reviewer retains their original score because the central "episodic" framing remains conceptually contentious due to the lack of an explicit, active memory module to support the claimed "mental time travel" across sequential experiences.


Reviewer FoJF (Final: 4): The rating remains unchanged as the reviewer continues to perceive the technical novelty as limited, arguing that the multi-step prediction approach is not significantly distinguished from prior works like Seer.


Reviewer 1yXu (Final: 4): The reviewer stands by their initial assessment, maintaining that the architecture is an incremental extension of the VPP baseline and that its minor improvements on standard benchmarks do not justify the increased system complexity.

---

### Decision · Program_Chairs · 2026-01-26

Reject